# Scale-Invariance in AI Representation Predicts AI-Brain Alignment

## Abstract

Understanding why some neural network representations align better with brain activity is essential for uncovering neural coding principles and developing human-like AI. While prior work has largely focused on model-level factors, such as dataset scale and task design, we focus on the rarely explored, yet more in-depth embedding level. Motivated by evidence that scale-invariance is widespread in biological neural systems, we identify it as a key embedding-level property. Analyzing 60 pretrained visual models and fMRI responses to natural images, we find that embeddings with stronger scale-invariance align better with fMRI. Training strategies modulate scale-invariance, with larger pretraining datasets enhancing it and fine-tuning reducing it, thereby affecting alignment performance. These findings establish scale-invariance as a fundamental embedding-level property that links training strategies to brain-like representations and suggest its potential as a guiding principle for designing more human-like AI.

## 1 Introduction

Recent studies show that neural networks, although never trained on neural data, often align well with brain activities (Yamins et al., 2014; Khaligh-Razavi & Kriegeskorte, 2014; Güçlü & Van Gerven, 2015; Eickenberg et al., 2017; Wang et al., 2023; Conwell et al., 2024; Shen et al., 2024; Raugel et al., 2025). This suggests that such networks may have developed representational structures similar to those in the brain. A question is therefore what kinds of model representations align better with brain activities. Studying this question can help improve model design to make them more brain-like. Such models can then serve as proxies for the brain, allowing us to analyze how information is progressively transformed into internal representations, thereby offering new insights into the neural coding mechanisms of the brain (Liu et al., 2025; Kanwisher et al., 2023).

There are two ways to explain why some models align better with brain activities. One is at the **model level**, focusing on training configurations such as dataset scale or task design. The other is at the **embedding level**, where alignment is explained through intrinsic properties of the learned representations. Most existing work has concentrated on the first perspective, showing that models trained on larger datasets or with self-supervised learning task achieve stronger alignment with brain activities (Wang et al., 2023; Conwell et al., 2024; Raugel et al., 2025). However, these findings remain at the level of training setups and do not clarify what representational changes actually support improved alignment. To make progress, *it is important to identify embedding-level factors that directly capture representational geometry and can quantitatively predict alignment performance* (Figure 1A).

Embedding-level factors include a wide range of properties, such as dimensionality, curvature or manifold smoothness. Since our focus is on AI-Brain alignment, we seek factors that are both consistently observed in biological neural systems and computable in artificial networks. Motivated by evidence that neural representations often exhibit scale-invariant patterns across space and time (He, 2014; Eguiluz et al., 2005; He et al., 2010; Gauthaman et al., 2024), we select **scale invariance** as a key candidate for analysis.

To quantify scale-invariance and assess its relationship with alignment performance, we introduce two complementary embedding-level metrics: **multi-scale dimensional stability**, which captures the stability of representational complexity at different scales, and **multi-scale distributional similarity**, which evaluates how well the overall structure is preserved across scales. This design is

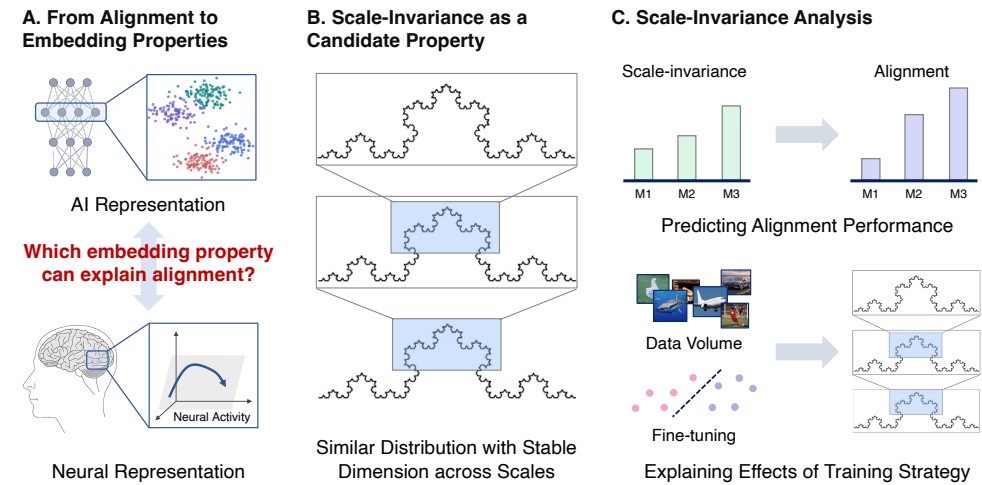

Figure 1: **Scale-Invariance Links Neural and AI Representations.** (A) Identifying embedding-level geometric properties provides a mechanistic explanation for AI-brain alignment. (B) Scale-invariance, a pervasive phenomenon in neural systems, motivates its study as a candidate property. Fractals provide a canonical example of scale-invariant structures, exhibiting similar patterns and preserved details across scales. (C) Scale-invariance analysis effectively predicts AI-brain alignment and explains the effects of pretraining dataset size and fine-tuning on alignment performance.

motivated by the intuition that a scale-invariant representation preserves both global shape and detail when zoomed in or out, much like a fractal pattern (Mandelbrot, 1989) (Figure 1B). Together, these metrics provide a rigorous and complementary characterization of scale-invariance in learned embeddings.

Our analysis yields two main findings (Figure 1C). First, the proposed metrics of scale invariance reliably predict alignment performance, confirming their utility as embedding-level descriptors. Second, they reveal how different training strategies shape representational geometry: larger pretraining datasets increase scale invariance and thereby improve alignment, whereas fine-tuning reduces scale invariance and weakens alignment with brain activity.

Our contributions are threefold:

- We identify scale invariance as a key embedding-level property for explaining AI-brain alignment and propose complementary metrics to quantify it in learned representations.

- We show that these metrics reliably predict AI-brain alignment across a diverse set of pretrained visual models, establishing scale invariance as a robust embedding-level predictor.

- We reveal how training strategies systematically shape both scale invariance and AI-brain alignment: larger pretraining datasets enhance scale invariance and consequently improve alignment, whereas fine-tuning reduces scale invariance, leading to weaker alignment.

## 2 RELATED WORK

**Alignment between AI and Brain**  Neural networks, especially large-scale pretrained models, increasingly exhibit strong alignment with neural activity, particularly when trained on diverse datasets (Yamins et al., 2014; Khaligh-Razavi & Kriegeskorte, 2014; Güçlü & Van Gerven, 2015; Eickenberg et al., 2017; Wang et al., 2023; Conwell et al., 2024; Shen et al., 2024; Raugel et al., 2025). Beyond evaluation, such alignment provides insights into brain function (Waldrop, 2024). For example, comparing models trained on different tasks shows that models trained with an autoregressive language modelling objective align more closely with neural activities than those trained under alternative paradigms, suggesting that the brain may employ predictive-coding-like mechanisms in language processing (Schrimpf et al., 2021).

**Scale-Invariance in Neural Systems.** Scale-invariance is a fundamental property of neural systems, observable across structure, dynamics, and representations. Structurally, the brain displays self-similar patterns across scales, such as in cortical folding, dendritic branching, and large-scale network connectivity (Grosu et al., 2023; Wang et al., 2019; Liao et al., 2023). Dynamically, neural activity exhibits scale-free fluctuations, manifested in power-law distributions of neural avalanches and long-range temporal correlations (Beggs & Plenz, 2003; Klaus et al., 2011; Milton, 2012; Friedman & Landsberg, 2013; Hu et al., 2013; Palva & Palva, 2014; Schaworonkow et al., 2015; Linkenkaer-Hansen et al., 2001), indicating that neural processes unfold similarly across timescales. At the level of neural representations, recent studies show that either the covariance spectra or the decay of variance across principal component dimensions often follows a power-law (Gauthaman et al., 2024; Stringer et al., 2019), suggesting that scale-invariant structure extends to how information is represented in the brain. These converging lines of evidence imply that scale-invariance is a unifying principle shaping neural architecture, activity, and computation.

**Methods for Analyzing Scale-Invariance.** Currently, the most widely used approaches for analyzing scale-invariance rely on identifying power-law relationships. Typical examples include analyzing the eigenvalue spectrum of covariance matrices or examining the decay of variance across principal component dimensions, which often follows a power-law (Gisiger, 2001; Stanley et al., 2000; Stringer et al., 2019; Gauthaman et al., 2024; Wang et al., 2025). While these methods are effective at detecting whether a system exhibits scale-invariant behavior, they provide limited insight into the degree to which a system approaches true scale-invariance. Consequently, new frameworks are needed that can quantitatively assess the proximity to scale-invariance and characterize multi-scale structure in high-dimensional embeddings.

## 3 PRELIMINARIES AND TECHNICAL BACKGROUND

This section introduces the concepts and methods used to define and measure scale invariance for embeddings, and it summarizes the data and alignment pipeline used in our experiments. We begin with a concise, general definition of scale invariance in section 3.1, then describe how to operationalize the notion of scale for discrete embedding sets in section 3.2. Section 3.3 presents two complementary structural properties that capture scale invariance, followed by their quantification in section 3.4. Finally, section 3.5 introduces the datasets and fMRI-to-model alignment procedure used to evaluate representational similarity.

### 3.1 GENERAL NOTION OF SCALE INVARIANCE

In mathematics, physics and statistics, scale invariance refers to the notion that a rule or relationship remains unchanged across scales. A common functional formulation is that a function $f$ satisfies

$$f(\lambda x) = \lambda^\alpha f(x) \quad \text{for all } \lambda > 0,$$

for some exponent $\alpha$. This equation means that scaling the input by a factor $\lambda$ leads to a predictable scaling of the output by $\lambda^\alpha$, so relative relations specified by $f$ are preserved across scales.

In the context of embeddings, we are interested in whether the geometric and relational structure of a set of representations remains consistent when moving from local to progressively larger neighborhoods. This perspective motivates our multi-scale analysis framework.

### 3.2 SCALE FOR EMBEDDING SETS

We first describe how to define scale for embedding sets (Figure 2A). Let $Z = \{z_i\}_{i=1}^N \subset \mathbb{R}^d$ be a set of embeddings. We treat an integer $K \in \{1, \ldots, N\}$ as the scale parameter. For a randomly selected anchor $z \in Z$, the set of points at scale $K$ is the $K$ nearest neighbors of $z$ in $Z$, denoted $\mathcal{N}_K(z)$. Small values of $K$ probe local structure around the anchor, capturing information from nearby points. Larger values of $K$ incorporate progressively more global structure, meaning that we start to include points that are farther away, revealing broader patterns and relationships. To put it simply, as we increase $K$, we're zooming out and looking at the neighborhood around the anchor from a wider perspective. In all analyses, we vary $K$ over a prescribed range of scales and report how statistics evolve with $K$.

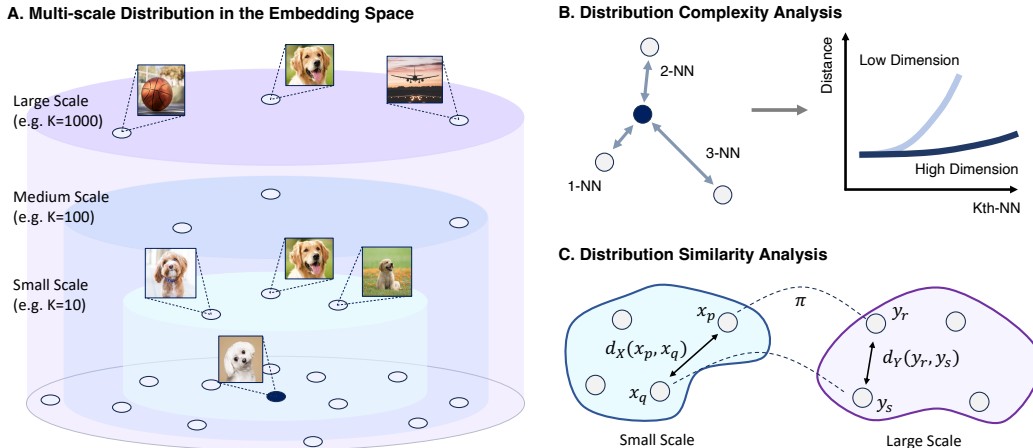

Figure 2: **Quantifying Scale-Invariance via Multi-Scale Geometry and Distribution.** **(A)** We define scale by varying neighborhood size: smaller scales correspond to smaller local neighborhoods. Based on this definition, we evaluate both the intrinsic dimensionality of embeddings at each scale and the similarity of distributions across scales. **(B)** Dimensionality is estimated from the growth rate of neighbor distances as neighborhood size increases: faster growth indicates lower intrinsic dimension. Performing this estimation at every scale allows us to track how dimensionality changes with scale. **(C)** Distributional similarity across scales is assessed using the Gromov–Wasserstein distance, which quantifies how consistent local neighborhood structures remain as scale increases.

### 3.3 TWO COMPLEMENTARY STRUCTURAL PROPERTIES

We quantify approximate scale invariance using two complementary, empirically measurable quantities evaluated as functions of the scale parameter $K$:

- **Multi-scale dimensional stability.** This quantity measures how the estimated intrinsic dimension $\bar{m}(K)$ changes as $K$ varies. A roughly constant $\bar{m}(K)$ indicates that the geometric complexity of the embedding remains similar across scales. In other words, we can characterize the distribution at different scales using a unified measure of complexity.

- **Multi-scale distributional similarity.** This quantity measures the similarity between embedding subsets at different scales, capturing how much the overall shape or distribution of the embedding is preserved when considering neighborhoods of size $K$.

Both properties are computed as functions of $K$ so that we can quantify the degree to which an embedding approximates geometric scale invariance.

### 3.4 QUANTIFICATION OF SCALE INVARIANCE

**Dimensional stability.** For an anchor $z$ and neighborhood size $K$ we estimate the local intrinsic dimension with the maximum likelihood estimator (Levina & Bickel, 2004)

$$\hat{m}_K(z) \;=\; \left[ \frac{1}{K-1} \sum_{j=1}^{K-1} \log\left( \frac{T_K(z)}{T_j(z)} \right) \right]^{-1},$$

where $T_j(z)$ is the Euclidean distance from $z$ to its $j$-th nearest neighbor in $Z$. The intrinsic dimension is estimated for each sample $z$, and these estimates are averaged to yield the final mean intrinsic dimension $\bar{m}(K)$.

Intuitively, $\bar{m}(K)$ captures how local complexity evolves as we move from nearest neighbors to more distant ones. In high-dimensional spaces, points tend to be closer together, so distances to successive neighbors increase slowly. In low-dimensional spaces, points are more spread out, and these distances grow more rapidly (Figure 2B).

To quantify dimensional stability across scales, we compute $\bar{m}(K)$ for a range of neighborhood sizes $K$, effectively estimating the intrinsic dimension at multiple scales. We then fit a linear regression of $\bar{m}(K)$ against $K$, and use the regression **slope** as a stability metric. Slopes close to zero indicate that the intrinsic dimension remains relatively constant across scales, reflecting high dimensional stability. Algorithmic implementation details are provided in Appendix C.

**Multi-scale distributional similarity.** Using the embeddings obtained at different scales as defined in Section 3.2, we randomly sample two subsets of points with the same number of samples but corresponding to different scales. Our aim is to quantify the similarity between the distributions of these two embedding subsets.

A natural metric for this purpose is the **Gromov–Wasserstein (GW) distance**, which measures the discrepancy between the relational geometries of two point sets (Figure 2C). GW distance is particularly suitable here because it does not require exact point correspondences and is invariant to rotations or reindexing of points, making it robust to differences in scale and sampling.

For a fixed anchor point $z$ and neighborhood sizes $K_a$ and $K_b$, we define the metric–measure spaces

$$X = (\mathcal{N}_{K_a}(z), d_X, \mu_X), \qquad Y = (\mathcal{N}_{K_b}(z), d_Y, \mu_Y),$$

where $\mathcal{N}_K(z)$ denotes the $K$-nearest neighbors of $z$, $d_X$ and $d_Y$ are pairwise Euclidean distances within each neighborhood, and $\mu_X, \mu_Y$ are uniform probability measures. The squared GW distance

$$\mathrm{GW}^2(X,Y) \;=\; \min_{\pi \in \Pi(\mu_X, \mu_Y)} \sum_{p,q,r,s} \left| d_X(x_p, x_q) - d_Y(y_r, y_s) \right|^2 \pi_{pr} \pi_{qs}$$

quantifies the minimal distortion needed to align the pairwise distance structures of $X$ and $Y$. Here, $\pi$ is a coupling matrix assigning soft correspondences between points in $X$ and $Y$, the minimization identifies the matching that best preserves relational geometry.

Intuitively, a smaller GW distance indicates that the overall geometric arrangement of points is largely preserved across scales. To assess multi-scale stability, we compute $\mathrm{GW}(\mathcal{N}_{K_0}(z), \mathcal{N}_K(z))$ with a fixed reference size $K_0 = 1000$ and varying $K$ from 1500 to 10000. Lower GW values indicate stronger preservation of relational structure as neighborhood size increases. We summarize these results using two statistics:

1. the **mean GW** across all tested scales, reflecting the average distributional similarity between the reference and larger-scale embeddings; and

2. the **regression slope of GW on** $K$ (from 1500 to 10000), capturing the rate at which similarity changes with scale.

Conceptually, this approach quantifies whether the local relational structure of the embedding "stretches" consistently across scales, providing a multi-scale measure of geometric stability. The full pairwise GW matrix across all examined $(K_a, K_b)$ is provided in Appendix F.2.

### 3.5 Alignment with fMRI responses

We evaluate the relationship between geometric scale invariance and neural alignment using the Natural Scenes Dataset (NSD) (Allen et al., 2022). NSD is a high-resolution 7T fMRI dataset collected from 8 subjects while they viewed large sets of natural images. For each image we extract model embeddings from a diverse collection of 60 pretrained vision models. We then follow a well-established and widely used analysis pipeline in the neuroscience and vision communities, predicting voxel-wise responses from model embeddings using ridge regression applied independently to each voxel, with 80% of the data used for training and the remaining 20% held out for evaluation (Yamins et al., 2014; Khaligh-Razavi & Kriegeskorte, 2014; Cichy & Kaiser, 2019; Conwell et al., 2024). See Appendix A for additional details. Alignment quality is summarized on the held-out set by the coefficient of determination ($R^2$), which we refer to as the alignment score. Higher alignment scores indicate better correspondence between the embeddings and fMRI. Model architecture details are provided in Appendix B.

# 4 RESULTS

## 4.1 ALIGNMENT PERFORMANCE ACROSS REGIONS

We first evaluated alignment between pretrained embeddings and neural responses across the whole cortex. This analysis provides an overview of regional alignment perdormance and highlights variability across models.

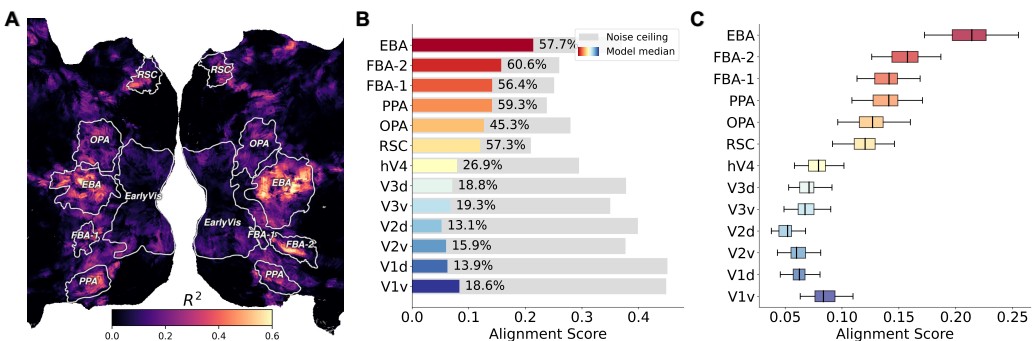

Figure 3: **AI-Brain Alignment Highlights Visual Cortex as a Benchmark. (A)** Whole-brain alignment maps show strongest alignment in visual regions, with high-level visual areas outperforming early visual areas. **(B)** Alignment scores normalized by noise ceiling reveal that regions like EBA reach ∼60% of ceiling, indicating substantial alignment. **(C)** Even in best-fitting regions, alignment varies widely across models.

Figure 3A shows that alignment is strongest in the visual cortex. Among higher-level visual regions such as EBA, FBA and PPA, alignment reaches approximately 60% of the noise ceiling (Figure 3B), indicating that pretrained embeddings account for a substantial fraction of stimulus-driven neural variance. Here, the *noise ceiling* represents the upper bound on explainable variance, determined by trial-to-trial variability in neural responses to repeated presentations of the same stimulus. Details are provided in Appendix A.

Model-to-model variability is also pronounced. For example, in EBA the best-performing model reaches 68.8% of the noise ceiling, while the weakest achieves only 46.5% (Figure 3C). Because EBA combines high overall alignment with large between-model variability, it offers a sensitive testbed for probing which embedding properties predict neural alignment. We therefore focus subsequent analyses on EBA.

Full results for whole-brain alignment, as well as results for other subjects, are reported in Appendix D.

## 4.2 MULTI-SCALE DIMENSIONAL STABILITY PREDICTS NEURAL ALIGNMENT

We next asked whether the stability of intrinsic dimensionality across scales explains differences in neural alignment. For each model, we estimated intrinsic dimensionality at multiple scales (Figure 4A) and summarized its stability by the regression slope of dimensionality versus $K$.

Results show that models with more stable dimensionality (i.e., slopes closer to zero) consistently achieve stronger alignment with neural responses (Figure 4B). Result of other regions, including both early visual areas (V1V4) and higher-level regions such as PPA, are reported in Appendix E.

## 4.3 MULTI-SCALE DISTRIBUTION SIMILARITY PREDICTS NEURAL ALIGNMENT

We then tested whether multi-scale distribution similarity predicts alignment. For each model, we computed GW distances between local neighborhoods at a fixed reference scale (K=1000) and neighborhoods at progressively larger scales (from K=1500 to K=10000) (Figure 5A). We summarized the results using both the mean GW distance and the regression slope of GW distance as scale increases, which captures the extent to which distributional similarity changes across scales.

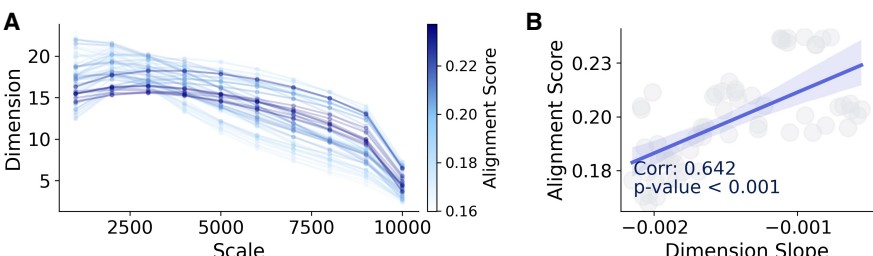

Figure 4: **Stable Intrinsic Dimension Across Scales Predicts Better Alignment. (A)** Models with higher alignment show stable dimensionality across scales. **(B)** Alignment score grows as the slope of dimension versus scale approaches zero, suggesting that scale-invariant geometry facilitates brain-like representations.

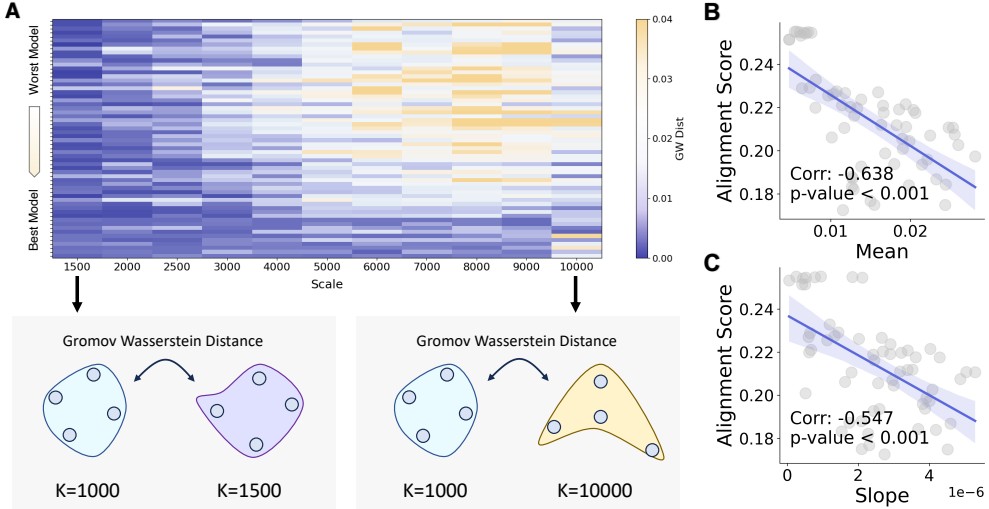

Figure 5: **Distributional Similarity Across Scales Tracks Alignment. (A)** Using the smallest scale as reference, models with higher alignment maintain higher distribution similarity across larger scales. **(B)** Lower mean Gromov-Wasserstein distance predicts stronger alignment. **(C)** Flatter GW slopes across scales indicate more stable GW distance across scales, further correlating with alignment.

Results show that models with lower mean GW distance, indicating higher multi-scale distribution similarity, consistently achieve stronger alignment with neural responses (Figure 5B). Alignment is also stronger for models with flatter GW slopes (closer to 0), reflecting more stable distributional similarity across scales (Figure 5C).

Together, these findings demonstrate that both the overall level of multi-scale distribution similarity and its stability across scales are predictive of neural alignment. A full analysis comparing all scale pairs yields the same consistent pattern (Appendix F).

## 4.4 TRAINING DATA VOLUME STRENGTHENS SCALE-INVARIANCE

Larger pretraining datasets lead to improved alignment with neural activity (Wang et al., 2023; Conwell et al., 2024). To test whether this effect is mediated by scale-invariance, we compared model variants trained on ImageNet-1K, ImageNet-12K and ImageNet-22K. In Figure 6A, points connected by lines represent matched variants within the same model family (for example small, medium and large versions of a given architecture).

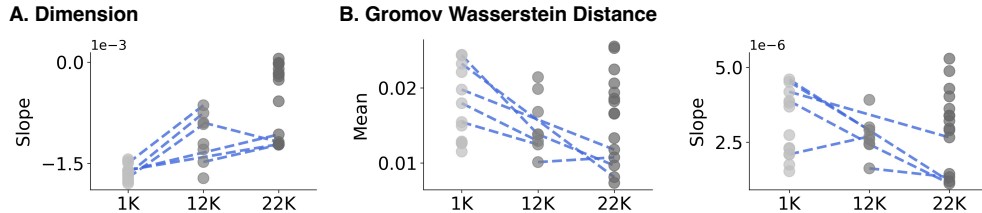

Figure 6: **Pretraining Data Scale Enhances Scale-Invariance. (A)** Larger datasets systematically strengthen multi-scale dimensional stability. **(B)** Larger datasets also increase multi-scale distributional similarity. Together, these results suggest that data abundance not only improves task performance but also fosters scale-invariant geometry, promoting AI-brain alignment.

Across matched variants, larger pretraining datasets consistently produce stronger dimensional stability (slope closer to 0; Figure 6A) and higher multi-scale distribution similarity (Figure 6B). This pattern indicates that scale-invariance itself grows with dataset size, providing a plausible mechanism for the improved neural alignment observed after large-scale pretraining.

Larger pretraining datasets are often accompanied by larger model sizes due to scaling laws. To examine the relationship between model size and scale-invariance, we performed a correlation analysis reported in Appendix G. We found that dimensional stability is significantly correlated with the number of model parameters (Pearson $r = 0.415$, $p < 0.05$), whereas multi-scale distribution similarity shows no significant correlation. Furthermore, in a multiple regression analysis, adding scale-invariance metrics on top of model size substantially improves the prediction of neural alignment ($R^2$ increases significantly compared to using model size alone), indicating that scale-invariance provides explanatory power for alignment performance beyond model size, and is not merely a reflection of model parameters.

### 4.5 FINE-TUNING DISRUPTS SCALE-INVARIANCE AND REDUCES ALIGNMENT

Previous work has shown that self-supervised pretraining produces embeddings with stronger neural alignment than standard supervised training (Conwell et al., 2024). Building on this, we asked whether fine-tuning preserves or disrupts the scale-invariant structure that supports neural alignment. Specifically, we directly compared pretrained models before and after fine-tuning to assess (i) whether fine-tuning alters alignment with neural responses, and (ii) how such changes relate to the multi-scale geometric properties of the embeddings. This analysis allows us to probe how task-specific adaptation impacts the structural features of embeddings that underlie their correspondence with neural representations.

Fine-tuned embeddings exhibit a significant reduction in alignment performance (Figure 7A). Both dimensional stability and multi-scale distributional similarity are decreased following fine-tuning (Figure 7B and C). We interpret this as fine-tuning reshaping embeddings into label-centered clusters (Figure 7D). As the scale increases, more clusters are included, amplifying differences across scales (Figure 7E and F).

To quantify this effect, we computed Silhouette Scores across scales (Figure 7G). Silhouette Scores measure how well data points cluster together: higher scores indicate that points are closer to others in the same class than to points in different classes. Pretrained embeddings without fine tuning show low, nearly scale-invariant scores, reflecting weak clustering across scales. In contrast, fine-tuned embeddings exhibit strong clustering at small scales, but the scores decline as neighborhood size grows. This decline reflects a dilution of class structure: at large scales, many classes contribute only a few samples each, making clusters appear weaker even if local class grouping is preserved.

Together, these results indicate that fine-tuning drives embeddings into class-centric clusters, disrupting multi-scale distributional similarity and thereby weakening neural alignment.

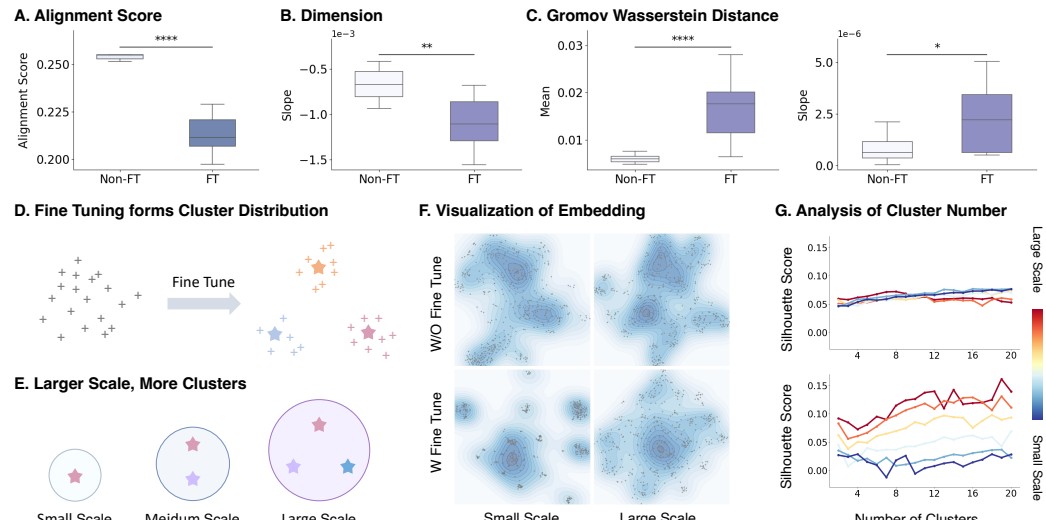

Figure 7: **Fine-Tuning Disrupts Scale-Invariance and Reduces Alignment. (A)** Alignment scores drop after fine-tuning. **(B)** Fine-tuning reduces multi-scale dimensional stability. **(C)** It also decreases multi-scale distributional similarity. **(D)** Embeddings become clustered by labels after fine tuning, **(E)** with larger scales capturing more clusters and amplifying scale differences. **(F)** Visualizations show stronger clustering in fine-tuned embeddings across scales. **(G)** Silhouette scores confirm that fine-tuning increases scale-dependent clustering, jointly explaining reduced alignment.

## 5 CONCLUSION

In this work, we investigated the role of **scale-invariance in embedding spaces as a predictor of AI-brain alignment**. By quantifying multi-scale dimensional stability and distributional similarity, we demonstrated that embeddings with stronger scale-invariance consistently align more closely with brain activity. Furthermore, we showed that different training strategies systematically shape scale-invariance: pretraining on larger datasets enhances it, thereby improving alignment, while fine-tuning reshapes embeddings into class-centered clusters, reducing scale-invariance and weakening alignment.

We further validated our results on the ResNet architecture, confirming that our findings are robust and consistent across different model architectures (Appendix I). Following the conventional approach, we also analyzed embeddings' power-law behavior. While this captures certain aspects of information organization, we found that its predictive power for alignment is weaker than our geometric scale-invariance metrics (Appendix L). Additionally, we manipulated the singular value distributions of embeddings to more closely follow a power-law, observing improved alignment, primarily driven by enhanced multi-scale dimensional stability (Appendix O).

Beyond identifying predictive markers, our framework provides a rigorous embedding-level perspective on representational geometry. Prior studies have highlighted factors such as pretraining data volume and task design (Wang et al., 2023; Conwell et al., 2024), but these operate at a coarse level and do not provide a direct, actionable approach for model design. In contrast, our metrics are fully computable and can be formulated as optimization objectives. For example, multi-scale distributional similarity, quantified using the Gromov-Wasserstein distance, is both tractable and suitable as a training regularizer. This opens the possibility of directly guiding models toward brain-like embeddings, reducing reliance on image-fMRI alignment data, and promoting data-efficient learning.

We also applied our analysis to fMRI (Appendix H). Compared to AI embeddings, neural representations in specific cortical regions (e.g., EBA, PPA) generally have lower dimensionality and show larger variations in both dimensionality and distributional similarity across scales. We interpret these observations as a consequence of the brains distributed coding: each region captures only a subset of stimulus information, whereas AI embeddings reflect a more complete representation of the input (Rissman & Wagner, 2012; Wen et al., 2024; Shin et al., 2021). As a result, scale-invariance

is less pronounced at the level of specific regions. Future work that integrates activity across multiple cortical areas to reconstruct more complete stimulus representations could enable more accurate assessments of scale-invariance in neural activity.

**Limitations.** A key limitation of our study is the difficulty of fully disentangling dataset size from model size, since scaling laws typically pair larger datasets with larger models (Kaplan et al., 2020). While our within-family comparisons partially mitigate this issue, residual correlations remain.

**Future Work.** Our results suggest two promising directions. First, it is important to test whether scale-invariance generalizes across modalities, for instance by examining language model embeddings and corresponding neural responses during reading comprehension. Second, incorporating scale-invariance as a regularization objective during pretraining may improve data efficiency, promote brain-like representations, and enhance generalization across tasks and domains.

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

## A    DETAILS OF FMRI ALIGNMENT

**Embedding extraction.**    For each image stimulus, we extract visual embeddings from pretrained vision models. All embeddings are projected to 300 dimensions using principal component analysis (PCA) for standardization across models and computational efficiency.

**Voxel-wise encoding model.**    Let $z_i \in \mathbb{R}^{300}$ be the embedding of image $i$, and let $r_{ij}$ be the fMRI response of voxel $j$ to image $i$. We train a voxel-specific ridge regression model:

$$r_{ij} \approx \mathbf{w}_j^\top \mathbf{z}_i + \epsilon. \tag{1}$$

Model fitting is done using 5-fold cross-validation on an 80/20 training/test split. The regularization parameter is tuned via nested cross-validation.

**Evaluation metrics.**    Alignment quality is quantified by the coefficient of determination ($R^2$) on held-out data.

**Noise ceiling estimation.**    Due to inherent measurement errors and individual variability in neural signals, even a perfectly accurate model cannot account for all sources of variation. Noise ceiling estimates are derived from the reliability of the beta weights across trials. In essence, the more consistent the neural response is across repeated presentations of the same image, the greater the proportion of the response variance that can be attributed to stimulus-driven signals. These estimates of the noise ceiling establish an upper bound on the amount of variance that can be explained or predicted in the response of a given voxel (Allen et al., 2022).

To quantify the noise ceiling in a more precise manner, the effective noise variance $N_{\text{eff}}$ is first computed by considering the number of times each image was presented to the subject, with different weights assigned to each trial type. Let $A$, $B$, and $C$ denote the number of distinct images presented three, two, and one times to the subject, respectively. The effective noise variance $N_{\text{eff}}$ is calculated as:

$$N_{\text{eff}} = \frac{\frac{A}{3} + \frac{B}{2} + \frac{C}{1}}{A + B + C}$$

Subsequently, the noise ceiling $N_C$ is determined by combining the signal variance $S^2$ with the effective noise variance $N_{\text{eff}}$, as follows:

$$N_C = \frac{S^2}{S^2 + N_{\text{eff}}}$$

Here, $S^2$ represents the signal variance, which is computed based on the beta weights derived from all NSD scan sessions. The data used is provided by the NSD dataset.

## B    MODEL ARCHITECTURE AND IMPLEMENTATION DETAILS

In this study, we employed the ConvNeXt architecture for generating embeddings across multiple models and datasets. In total, we utilized 60 ConvNeXt-based models under different paradigms, with their indices and training details provided in Table 1. Among them, the model indices are ordered according to their alignment scores with the fMRI signals in the EBA region, sorted from low to high.

ConvNeXt is constructed by progressively adapting the ResNet architecture to incorporate design principles from Vision Transformers, such as large kernel sizes, depthwise convolutions, and layer normalization (Liu et al., 2022). The architecture consists of four main stages with varying resolutions, each composed of multiple residual blocks. Key modifications in ConvNeXt include the replacement of ReLU with GELU activation for smoother gradients, the adoption of large kernel

sizes (e.g., 7x7) to increase receptive fields, the implementation of depthwise separable convolutions for efficient spatial mixing, the utilization of Layer Normalization instead of Batch Normalization, and the employment of a patchify stem to align with Vision Transformer preprocessing. ConvNeXt achieves competitive performance across multiple visual recognition tasks, surpassing the Swin Transformer in terms of accuracy and scalability while maintaining computational efficiency.

**Implementation via Huggingface API:** For this analysis, we leveraged the Huggingface Transformers library to access ConvNeXt models pre-trained on various datasets. Huggingface provides several ConvNeXt variants pre-trained on datasets such as ImageNet-1K and ImageNet-22K, allowing us to systematically compare model embeddings derived from different pre-training datasets while controlling for architectural variations.

**Computational Resources** We utilized an NVIDIA GeForce RTX 3080 GPU with 10GB GDDR6X memory, a 320-bit memory interface, and a memory bandwidth of 760 GB/s for embedding extraction using the Huggingface Transformers API. This setup adequately meets the computational requirements without requiring additional resources.

**Rationale for Using ConvNeXt:** The primary motivation for utilizing ConvNeXt in this study is its modular design, enabling consistent model architecture across multiple pre-training datasets. This setup allows for a controlled analysis of how pre-training data impacts alignment with neural data while minimizing confounding effects from architectural changes. Moreover, ConvNeXt's architectural simplicity and computational efficiency facilitate the generation of embeddings across multiple participants and brain regions, ensuring robust comparative analyses.

In summary, the ConvNeXt architecture serves as a robust and scalable backbone for examining the influence of pre-training datasets on model-brain alignment, providing a controlled framework for evaluating structural similarities between model embeddings and neural data.

Table 1: Model information including model index (sorted by alignment score in EBA), pretraining dataset, fine-tuning, and language modality usage.

| Model Name | Index | Pretraining Dataset | FT | Lang |
|---|---|---|---|---|
| convnext_nano.d1h_in1k | M1 | ImageNet-1K | ✗ | ✗ |
| convnext_nano_ols.d1h_in1k | M2 | ImageNet-1K | ✗ | ✗ |
| convnext_tiny_hnf.a2h_in1k | M3 | ImageNet-1K | ✗ | ✗ |
| convnext_pico_ols.d1_in1k | M4 | ImageNet-1K | ✗ | ✗ |
| convnext_pico.d1_in1k | M5 | ImageNet-1K | ✗ | ✗ |
| convnext_atto_ols.a2_in1k | M6 | ImageNet-1K | ✗ | ✗ |
| convnext_large.fb_in1k | M7 | ImageNet-1K | ✗ | ✗ |
| convnext_femto.d1_in1k | M8 | ImageNet-1K | ✗ | ✗ |
| convnext_base.fb_in1k | M9 | ImageNet-1K | ✗ | ✗ |
| convnext_atto.d2_in1k | M10 | ImageNet-1K | ✗ | ✗ |
| convnext_femto_ols.d1_in1k | M11 | ImageNet-1K | ✗ | ✗ |
| convnext_small.in12k_ft_in1k_384 | M12 | ImageNet-12K | ✓ | ✗ |
| convnext_small.in12k_ft_in1k | M13 | ImageNet-12K | ✓ | ✗ |
| convnext_small.fb_in1k | M14 | ImageNet-1K | ✗ | ✗ |
| convnext_tiny.fb_in1k | M15 | ImageNet-1K | ✗ | ✗ |
| convnext_base.clip_laion2b_augreg_ft_in1k | M16 | LAION-2B | ✓ | ✓ |
| convnext_nano.in12k_ft_in1k | M17 | ImageNet-12K | ✓ | ✗ |
| convnext_large_mlp.clip_laion2b_augreg_ft_in1k | M18 | LAION-2B | ✓ | ✓ |
| convnext_large_mlp.clip_laion2b_augreg_ft_in1k_384 | M19 | LAION-2B | ✓ | ✓ |
| convnext_tiny.in12k_ft_in1k | M20 | ImageNet-12K | ✓ | ✗ |
| convnext_base.clip_laiona_augreg_ft_in1k_384 | M21 | LAION-A | ✓ | ✓ |
| convnext_tiny.in12k_ft_in1k_384 | M22 | ImageNet-12K | ✓ | ✗ |
| convnext_tiny.fb_in22k_ft_in1k_384 | M23 | ImageNet-22K | ✓ | ✗ |
| convnext_tiny.fb_in22k_ft_in1k | M24 | ImageNet-22K | ✓ | ✗ |
| convnext_large_mlp.clip_laion2b_soup_ft_in12k_in1k_384 | M25 | LAION-2B | ✓ | ✓ |
| convnext_small.fb_in22k_ft_in1k_384 | M26 | ImageNet-22K | ✓ | ✗ |
| convnext_base.clip_laion2b_augreg_ft_in12k_in1k | M27 | LAION-2B | ✓ | ✓ |
| convnext_large_mlp.clip_laion2b_soup_ft_in12k_in1k_320 | M28 | LAION-2B | ✓ | ✓ |
| convnext_small.fb_in22k_ft_in1k | M29 | ImageNet-22K | ✓ | ✗ |
| convnext_base.clip_laion2b_augreg_ft_in12k_in1k_384 | M30 | LAION-2B | ✓ | ✓ |
| convnext_small.in12k | M31 | ImageNet-12K | ✗ | ✗ |
| convnext_base.fb_in22k_ft_in1k_384 | M32 | ImageNet-22K | ✓ | ✗ |
| convnext_large.fb_in22k_ft_in1k_384 | M33 | ImageNet-22K | ✓ | ✗ |
| convnext_xxlarge.clip_laion2b_soup_ft_in1k | M34 | LAION-2B | ✓ | ✓ |
| convnext_base.fb_in22k_ft_in1k | M35 | ImageNet-22K | ✓ | ✗ |
| convnext_xlarge.fb_in22k_ft_in1k_384 | M36 | ImageNet-22K | ✓ | ✗ |
| convnext_large_mlp.clip_laion2b_soup_ft_in12k_384 | M37 | LAION-2B | ✓ | ✓ |
| convnext_tiny.in12k | M38 | ImageNet-12K | ✗ | ✗ |
| convnext_large.fb_in22k_ft_in1k | M39 | ImageNet-22K | ✓ | ✗ |
| convnext_large_mlp.clip_laion2b_augreg_ft_in12k_384 | M40 | LAION-2B | ✓ | ✓ |
| convnext_xlarge.fb_in22k_ft_in1k | M41 | ImageNet-22K | ✓ | ✗ |
| convnext_nano.in12k | M42 | ImageNet-12K | ✗ | ✗ |
| convnext_small.fb_in22k | M43 | ImageNet-22K | ✗ | ✗ |
| convnext_base.clip_laion2b_augreg_ft_in12k | M44 | LAION-2B | ✓ | ✓ |
| convnext_large.fb_in22k | M45 | ImageNet-22K | ✗ | ✗ |
| convnext_xxlarge.clip_laion2b_soup_ft_in12k | M46 | LAION-2B | ✓ | ✓ |
| convnext_tiny.fb_in22k | M47 | ImageNet-22K | ✗ | ✗ |
| convnext_large_mlp.clip_laion2b_soup_ft_in12k_320 | M48 | LAION-2B | ✓ | ✓ |
| convnext_xlarge.fb_in22k | M49 | ImageNet-22K | ✗ | ✗ |
| convnext_base.fb_in22k | M50 | ImageNet-22K | ✗ | ✗ |
| convnext_xxlarge.clip_laion2b_rewind | M51 | LAION-2B | ✗ | ✓ |
| convnext_xxlarge.clip_laion2b_soup | M52 | LAION-2B | ✗ | ✓ |
| convnext_large_mlp.clip_laion2b_augreg | M53 | LAION-2B | ✗ | ✓ |
| convnext_large_mlp.clip_laion2b_ft_320 | M54 | LAION-2B | ✓ | ✓ |
| convnext_base.clip_laiona | M55 | LAION-A | ✗ | ✓ |
| convnext_large_mlp.clip_laion2b_ft_soup_320 | M56 | LAION-2B | ✓ | ✓ |
| convnext_base.clip_laion2b | M57 | LAION-2B | ✗ | ✓ |
| convnext_base.clip_laiona_320 | M58 | LAION-A | ✗ | ✓ |
| convnext_base.clip_laion2b_augreg | M59 | LAION-2B | ✗ | ✓ |
| convnext_base.clip_laiona_augreg_320 | M60 | LAION-A | ✗ | ✓ |

## C  BACKGROUND ON DIMENSIONAL STABILITY ANALYSIS

### C.1  FRACTAL DIMENSION AND CORRELATION DIMENSION

The concept of fractal dimension provides a robust framework for quantifying the complexity, ir-regularity and structural intricacies of datasets across different scales. Unlike traditional Euclidean dimensions, fractal dimensions capture the degree to which a set fills space as the observation scale varies. This is particularly useful in the context of high-dimensional data and manifold learning.

**Definition 1** *Fractal Dimension: The fractal dimension $D_f$ is defined as the scaling exponent that quantifies how the number of self-similar structures in a set changes with the scale of observation. Mathematically, it can be expressed as:*

$$D_f = \lim_{\epsilon \to 0} \frac{\log N(\epsilon)}{\log(1/\epsilon)}, \tag{2}$$

*where $N(\epsilon)$ represents the number of $\epsilon$-sized boxes required to cover the set.*

*A higher fractal dimension indicates a more intricate structure with greater self-similarity across multiple scales.*

### C.1.1 CORRELATION DIMENSION

The correlation dimension is a specific type of fractal dimension that focuses on the spatial distribution properties of a point set, effectively capturing statistical sparsity and clustering behavior. It provides a finer-grained analysis of the dataset structure, especially in cases where data points are non-uniformly distributed.

**Definition 2** *Correlation Dimension: The correlation dimension $D_C$ is defined based on the correlation function $C(\epsilon)$ as:*

$$D_C = \lim_{\epsilon \to 0} \frac{\log C(\epsilon)}{\log \epsilon}, \tag{3}$$

*where the correlation function $C(\epsilon)$ is given by:*

$$C(\epsilon) = \frac{1}{N(N-1)} \sum_{i=1}^{N} \sum_{j \neq i} \mathbb{I}(\|x_i - x_j\| < \epsilon). \tag{4}$$

- *$N$ is the number of data points.*

- *$\|x_i - x_j\|$ represents the Euclidean distance between points $x_i$ and $x_j$.*

- *$\mathbb{I}(\cdot)$ is the indicator function, equal to 1 if the condition inside is true and 0 otherwise.*

*The correlation dimension effectively estimates the likelihood that two randomly chosen points from the dataset are within a distance $\epsilon$ of each other, thus providing insights into the spatial organization of the data.*

### C.2 FRACTAL DIMENSION ESTIMATION USING MAXIMUM LIKELIHOOD

To accurately estimate the fractal dimension, particularly the correlation dimension, we employ the Maximum Likelihood Estimation (MLE) method as proposed by Levina and Bickel (Levina & Bickel, 2004). This method leverages distances between neighboring points to infer the intrinsic dimensionality of the underlying manifold.

### C.2.1 FRACTAL DIMENSION ESTIMATION VIA MLE

The MLE-based estimation method utilizes the $k$-nearest neighbor distances to quantify the dimensionality of a dataset. For a given data point $x_i$, the estimated fractal dimension $\hat{m}_k(x_i)$ at a specific $k$ value is defined as:

$$\hat{m}_k(x_i) = \left[ \frac{1}{k-1} \sum_{j=1}^{k-1} \log \frac{T_k(x_i)}{T_j(x_i)} \right]^{-1}, \tag{5}$$

where:

- $T_j(x_i)$ is the Euclidean distance from $x_i$ to its $j^{\text{th}}$ nearest neighbor.
- $k$ is the number of nearest neighbors considered for the estimation.

A smaller $k$ captures local structural variations, emphasizing finer-scale features, while a larger $k$ reflects broader, more global patterns in the data distribution.

### C.2.2 Averaging Over Data Points

To obtain a robust estimate of the fractal dimension for the entire dataset, we average the estimated dimensions across all data points:

$$\bar{m}_k = \frac{1}{N} \sum_{i=1}^{N} \hat{m}_k(x_i), \tag{6}$$

where $N$ is the total number of data points. This averaging process helps to mitigate the effects of noise and local variations, producing a stable estimate of the fractal dimension at a specific $k$ value.

### C.2.3 Selection of $k$ and Scale Analysis

The choice of $k$ significantly impacts the resulting dimension estimate:

- **Small** $k$: Focuses on finer-scale structures, capturing local density variations and small-scale clustering.

- **Large** $k$: Emphasizes broader trends, providing a more smoothed, global perspective of the dataset structure.

To achieve a comprehensive understanding of the manifold, it is essential to analyze the fractal dimension estimates across multiple $k$ values. This multi-scale analysis helps in identifying how the estimated dimensionality varies with scale, offering deeper insights into the intrinsic complexity of the dataset.

### C.3 Application to Manifold Complexity Analysis

Estimating the fractal and correlation dimensions enables us to quantify the intrinsic complexity of data manifolds, particularly in high-dimensional settings. By systematically analyzing the scaling behavior and spatial organization of the dataset across different $k$ values, we can effectively characterize manifold structures and their dimensional stability. This approach provides a robust framework for identifying structural patterns, assessing sparsity and detecting non-linear data distributions.

## D Supplementary Analysis on AI-Brain Alignment

To further investigate the voxel-wise alignment analysis presented in the main text, we extended the analysis across all eight participants using multiple models. Here, we focus on two key visualizations that highlight the variability and consistency of alignment performance.

Figure 8 illustrates the alignment performance of a selected model (Model index: M50) across the eight participants. Despite inter-subject variability, the overall spatial distribution of alignment remains relatively consistent across participants. Regions around the EBA consistently show higher alignment, indicating that model-derived representations align more effectively with neural activity in higher-order visual areas. This consistent spatial distribution suggests that the alignment patterns observed in the main analysis are not specific to a single participant but are generalizable across multiple subjects.

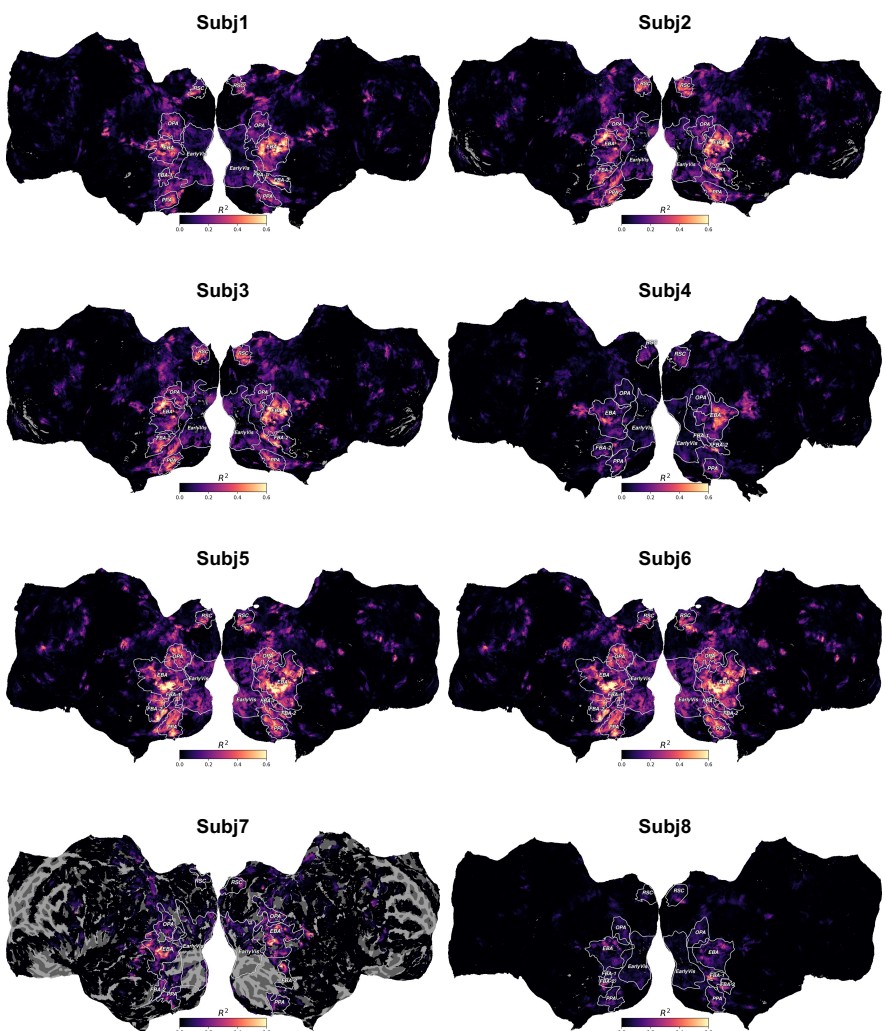

Figure 8: **Alignment performance across different subjects for the same model (Model index: M50).** This figure illustrates the alignment performance of the same model across different subjects. There is a noticeable variation in alignment effectiveness, with Subjects 5 and 6 exhibiting better alignment, while Subjects 7 and 8 show relatively weaker alignment. This highlights substantial inter-subject variability in alignment performance.

# E SUPPLEMENTARY ANALYSIS ON DIMENSIONALITY

We further extend the dimensionality analysis to encompass additional brain regions and participants, as illustrated in Figures 9 and 10. These figures depict the relationship between mean dimensionality and alignment scores, as well as the correlation between dimensional slope and alignment scores across various regions and subjects.

Consistent with the main text, the mean dimension provides limited predictive power, while the dimensional slope shows a consistently strong correlation with alignment, reaching up to 0.766 in RSC . This further supports the conclusion that dimensional stability across scales is a key factor in capturing brain-like representations.

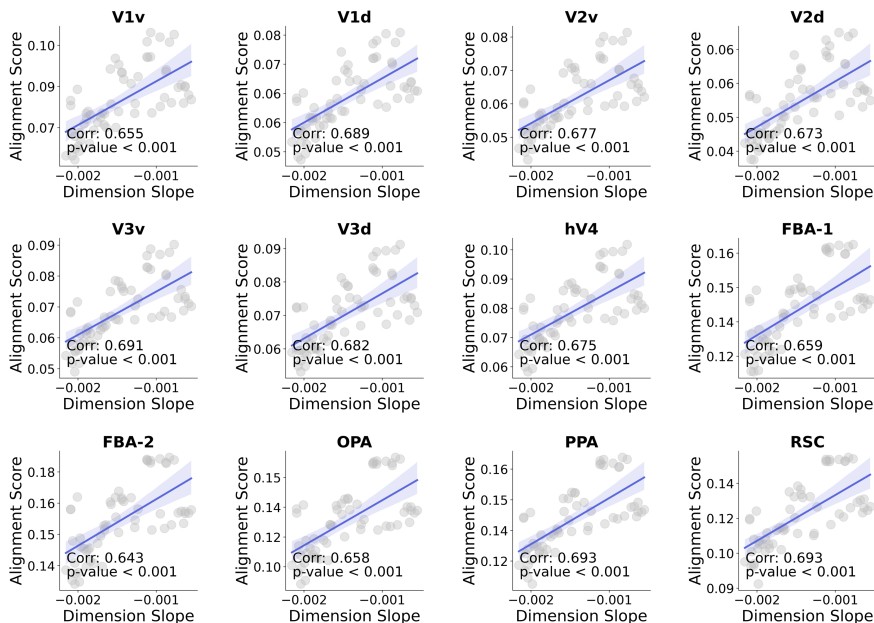

Figure 9: **Dimensional analysis across different ROIs within the same participant (Subject 1).** The dimensional slope are calculated for each ROI to assess how structural complexity varies across brain regions. The results reveals that the dimensional slope exhibits a strong negative correlation with alignment.

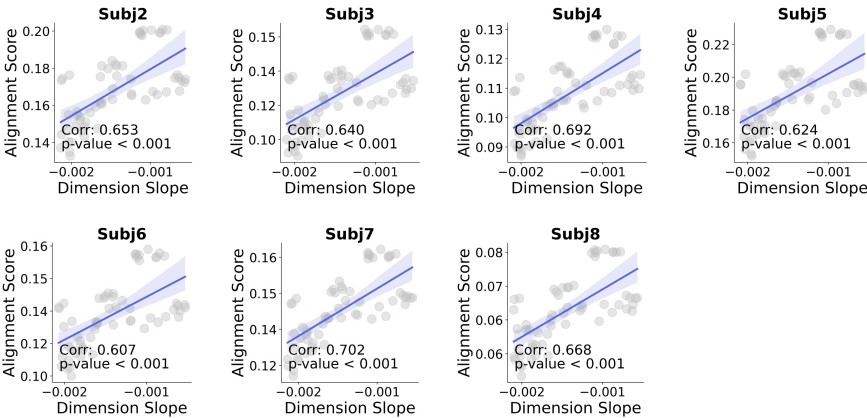

Figure 10: **Dimensional analysis across subjects for the same ROI (EBA).** This figure presents the dimensionality analysis for the embeddings within the EBA across multiple subjects. The dimensional slope is calculated based on the embeddings, remaining consistent across subjects, while alignment scores vary across individuals. The analysis highlights that the dimensional slope demonstrates a strong correlation with alignment scores, indicating that embeddings with more stable structural complexity across scales tend to achieve stronger alignment with neural data. This consistency in embedding structure, despite inter-subject variability in alignment scores, further underscores the importance of scale-invariant structure in capturing brain-like representations.

# F  SUPPLEMENTARY ON MULTI-SCALE DISTRIBUTION SIMILARITY ANALYSIS

## F.1  MULTI-SCALE DISTRIBUTION SIMILARITY ANALYSIS FOR MORE SUBJECTS AND REGIONS

In the main text, we focused on the Multi-scale Distribution Similarity Analysis within the EBA region. Here, we extend the analysis to additional brain regions and subjects to assess the generalizability of the observed patterns. By evaluating the Gromov Wasserstrin Distance across multiple regions and participants, we verify the consistency of the relationship between multi-scale distribution similarity and alignment performance. Figure 11 and Figure 12 confirm that embeddings with lower mean Gromov Wasserstrin Distance across scales and flatter slopes consistently align better with fMRI data across different cortical areas and subjects, reinforcing the importance of multi-scale distribution similarity as a robust predictor of alignment.

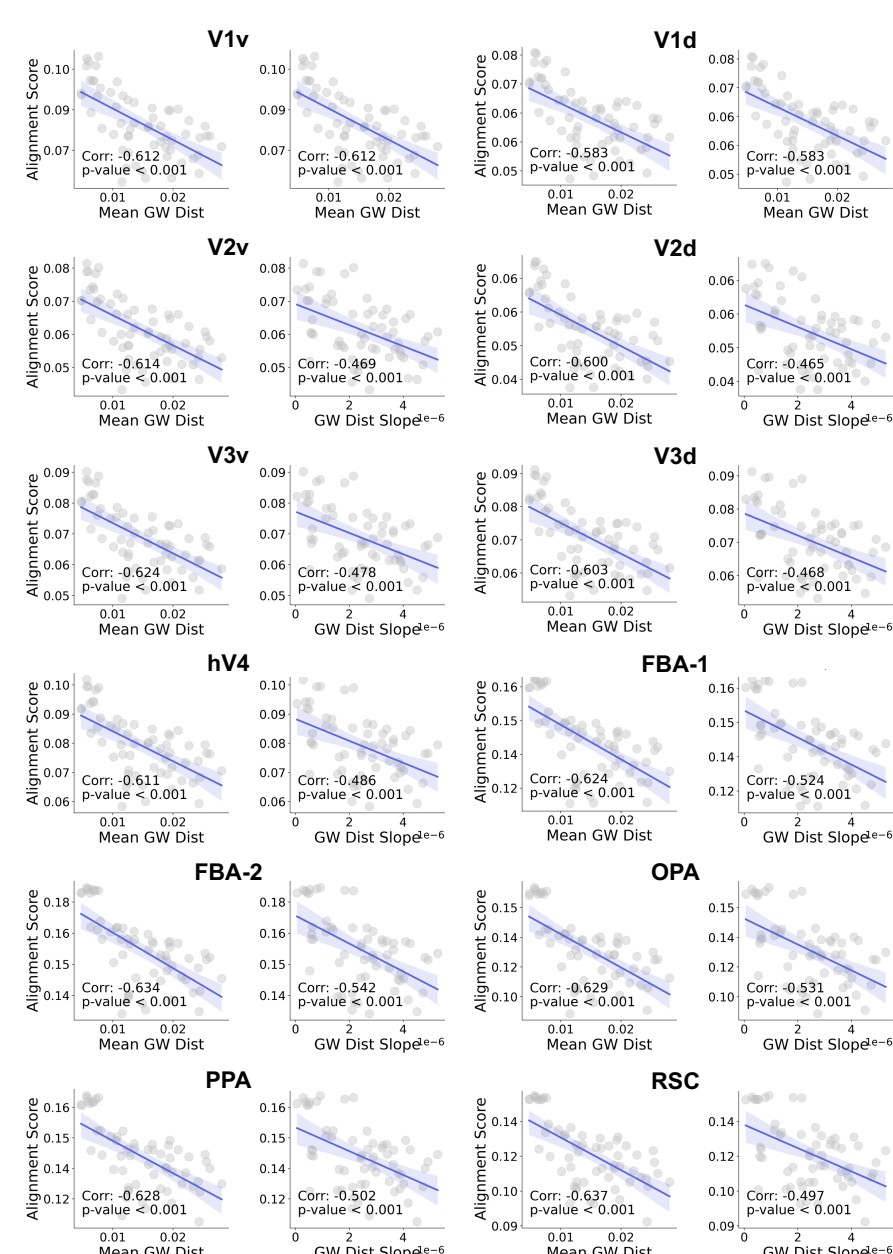

Figure 11: **Multi-scale distributional similarity across different ROIs within the same participant (Subject 1).** This figure examines the relationship between multi-scale distributional similarity and alignment performance across different brain regions for Subject 1. It can be observed that, across different ROIs within the same participant, higher distributional similarity consistently corresponds to higher alignment scores, thereby confirming the generalizability of this conclusion across ROIs.

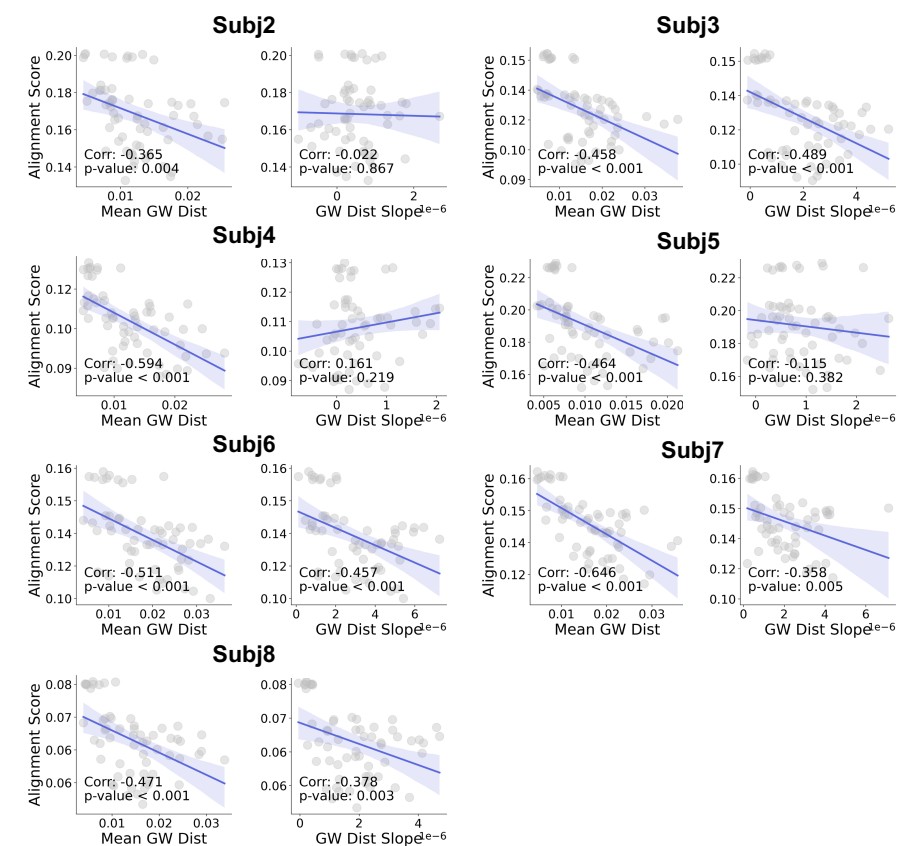

Figure 12: **Multi-scale distributional similarity across different ROIs within the same participant (Subject 1).** This figure shows the multi-scale distributional similarity analysis for the EBA region across multiple subjects. It can be observed that, for different participants, as distributional similarity increases, the alignment score also rises, thereby confirming the generalizability of this conclusion across subjects.

## F.2 MULTI-SCALE DISTRIBUTION SIMILARITY ANALYSIS FOR MORE SCALES

To comprehensively assess structural consistency across different embedding scales, we extended the structural similarity analysis to include all pairwise scale combinations, focusing specifically on the EBA region. This expanded analysis provides a more granular view of how structural organization evolves across scales and how it relates to alignment performance.

Figure 13 shows the r4esults of all 60 models, ordered by alignment performance from left to right. The plot reveals a consistent trend: as alignment improves, the multi-scale distribution similarity between different scales increases, evidenced by a reduction in gromov wasserstein distance.

To provide a more quantitative measure of structural variation, we calculated the variance of gromov wasserstein distance across all scale combinations. Figure 14 presents the variance analysis for all 60 models. The variance of gromov wasserstein distance systematically decreases as alignment improves, providing a more robust indicator of structural stability across scales. This finding underscores that models with stronger alignment not only exhibit lower gromov wasserstein distance but also maintain more consistent structural organization across varying scales, reflecting a more stable embedding space.

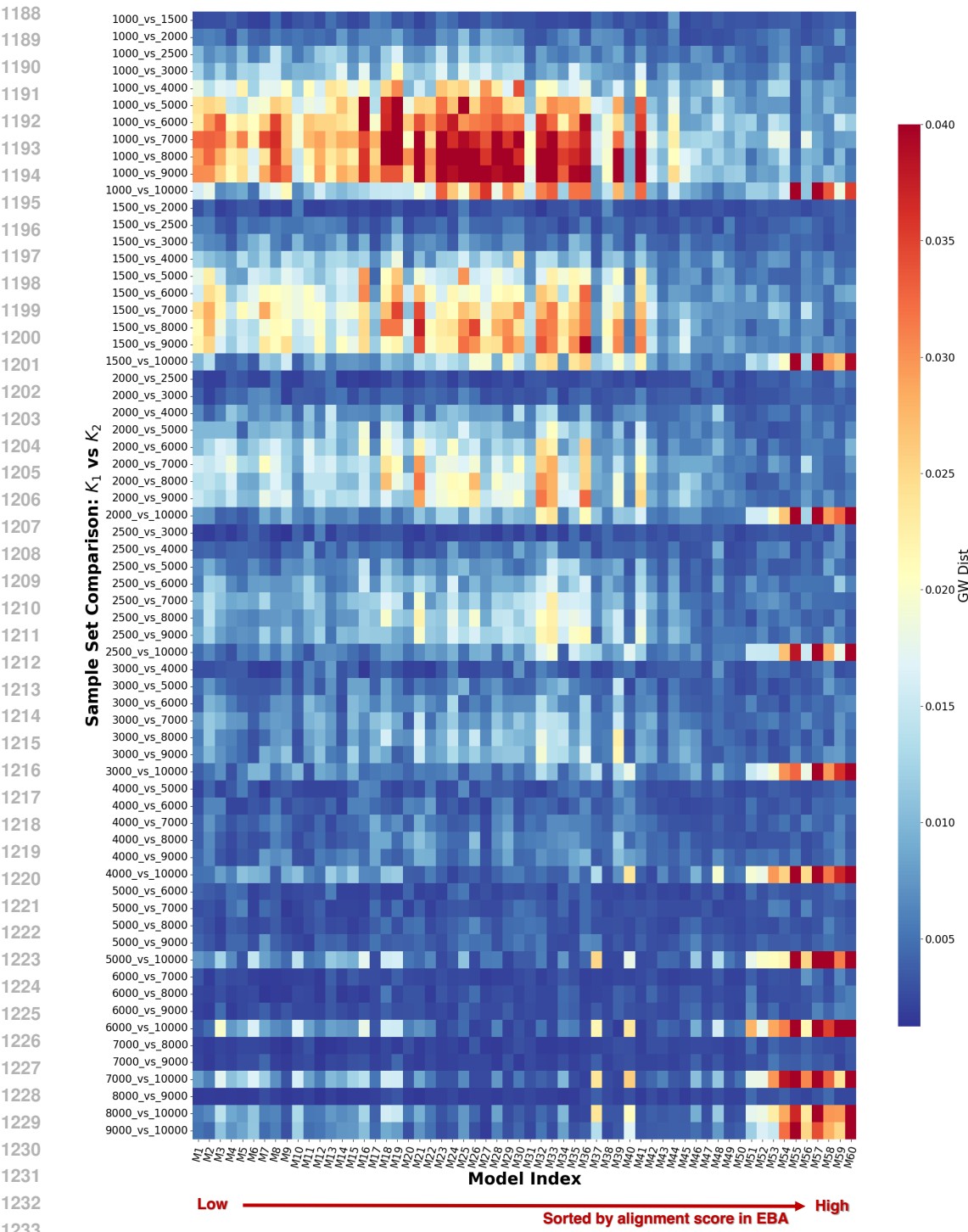

Figure 13: **Multi-scale distributional similarity for subj1 in EBA region.** The figure presents the gromov wasserstein distance for all 60 models, sorted by alignment performance from left (lowest alignment) to right (highest alignment). The analysis reveals a consistent trend where models with higher alignment exhibit lower gromov wasserstein distance.

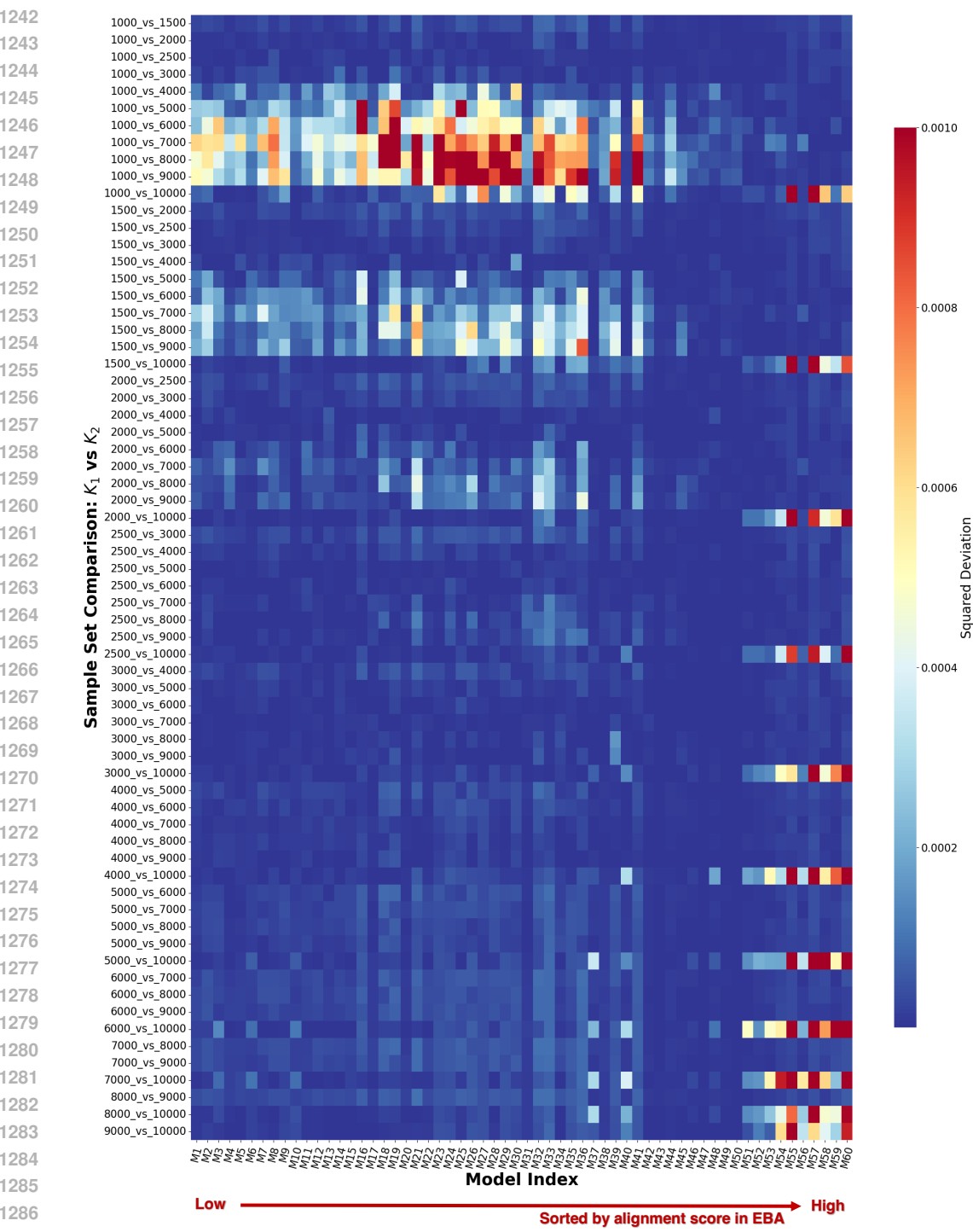

Figure 14: **Variance analysis of multi-scale distribution for subj1 in EBA region.** This figure presents the variance of multi-scale distribution similarity across all 60 models, systematically ordered by alignment performance from left (lowest alignment) to right (highest alignment). The analysis reveals a clear trend: as alignment performance improves, the variance of gromov wasserstein disance between different scales consistently decreases.

## G  RELATIONSHIP BETWEEN MODEL SIZE AND ALIGNMENT PERFORMANCE

In Section 4.4, we showed that larger pretraining datasets are associated with better alignment performance and stronger scale-invariance in embeddings. Since models trained on larger datasets often also have more parameters, it is natural to ask whether model size itself is significantly related to alignment performance, and whether embedding scale-invariance depends on the number of model parameters.

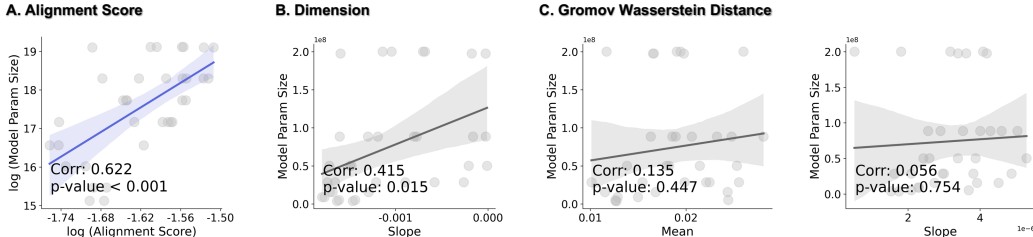

Figure 15: **Relationship between model parameter size and alignment performance. (A)** Alignment score versus model parameter size. **(B)** Dimensionality versus model parameter size. **(C)** Gromov-Wasserstein distance versus model parameter size.

As shown in Figure 15, model parameter size is indeed significantly correlated with alignment performance. However, we observe no significant relationship between parameter size and the scale-invariance of embeddings.

Table 2: Incremental contribution of scale-invariance metrics to alingment score fitting

| Variables | $R^2$ | Adj. $R^2$ | $\Delta R^2$ | $p$-value | Sig. |
|---|---|---|---|---|---|
| Baseline | 0.255 | 0.231 | – | – | – |
| Baseline + Dimension Slope | 0.816 | 0.804 | +0.561 | $< 0.001$ | *** |
| Baseline + Mean GW Dist | 0.270 | 0.223 | +0.015 | 0.007 | ** |
| Baseline + GW Dist Slope | 0.292 | 0.246 | +0.037 | 0.004 | ** |

We further constructed multiple regression models to predict alignment scores, using model parameter size and scale-invariance metrics as input features. Specifically, we first fit a baseline model with only parameter size as the predictor, and then extended this model by including a scale-invariance metric in addition to parameter size. Comparing these models, we found that adding the scale-invariance metric substantially increases the explained variance ($R^2$) in alignment scores (Table 2). This demonstrates that, although scale-invariance and model size are not directly correlated, they provide complementary information for predicting embedding alignment, with dimensional stability being particularly informative.

## H  SUPPLEMENTARY ANALYSIS ON SCALE-INVARIANCE IN FMRI DATA

In this section, we extend the scale-invariance analysis to the fMRI data from visual cortical regions, Subject 1, focusing on dimensional stability and multi-scale distribution similarity. For each region, voxel responses to 10000 visual stimuli were collected, yielding high-dimensional representations of size $(10000, \text{Voxel Num})$ for subsequent analysis.

Compared to AI embeddings, we observed that the dimensionality of fMRI representations is generally lower across most scales (Figure 16A). However, the dimensional stability of fMRI data is not higher than that of AI embeddings, and in some cases, it exhibits more variability. Similarly, multi-scale distribution similarity decreases substantially from small to large scales, indicating that fMRI are less structurally consistent across scales (Figure 16B).

We interpret these findings in terms of the brain's distributed representations. Each cortical region likely encodes only a subset of information, in contrast to neural network embeddings, where a

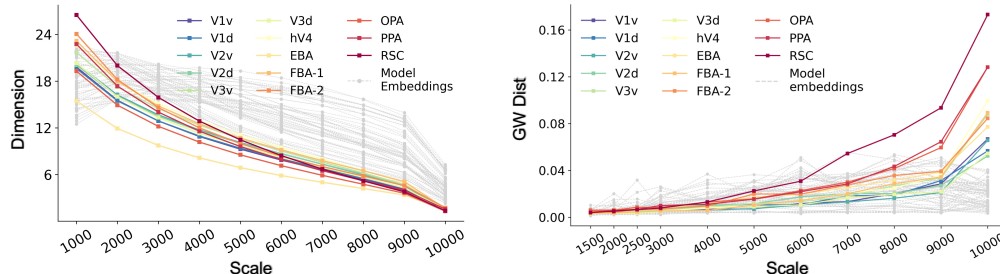

Figure 16: **Scale-Invariance Analysis of fMRI Data in Visual Cortex (Subject 1). (A)** Dimensionality across different scales. **(B)** Distribution similarity between the smallest scale and larger scales, measured by Gromov-Wasserstein distance.

single layer captures a more complete representation of the input data. In neural networks, layers with stronger scale-invariance tend to align better with fMRI responses. For the brain, however, extracting individual regions provides incomplete information. Analyzing all regions together would increase dimensionality but also introduce unrelated signals, complicating the analysis.

In summary, while fMRI data exhibit generally lower dimensionality than AI embeddings, their scale-invariance properties are limited, likely reflecting distributed, partial coding in the brain. This distributed nature makes direct comparison with AI embeddings challenging, particularly for interpreting scale-invariance and alignment.

# I   VALIDATION OF OTHER ARCHITECTURES

To evaluate the generalizability of the relationship between scale-invariance metrics and alignment performance, we extend our analysis beyond the ConvNeXT family to a broader set of architectures. In particular, we apply the same scale-invariance procedures to a diverse collection of ResNet models, including:

- **ResNet18 variants**: resnet18.a1_in1k, resnet18.a2_in1k, resnet18.a3_in1k
- **ResNet50 variants**: resnet50.a1_in1k, resnet50.a1h_in1k, resnet50.a2_in1k, resnet50.a3_in1k, resnet50.am_in1k
- **ResNet101 variants**: resnet101.a1_in1k, resnet101.a1h_in1k, resnet101.a2_in1k, resnet101.a3_in1k, resnet101.gluon_in1k
- **ResNet152 variants**: resnet152.a1_in1k, resnet152.a1h_in1k, resnet152.a2_in1k, resnet152.a3_in1k, resnet152.gluon_in1k

As shown in Figure 17, scale-invariance metrics robustly predict alignment performance across all tested architectures and brain regions (EBA, OPA and PPA). These results demonstrate that the observed relationship is not model-specific and further support the broad generalizability of our findings.

# J   INFLUENCE OF SCALE BOUNDARY

To assess the robustness of our dimensionality analysis, we examined the potential influence of boundary scales. First, we removed the largest scales (K = 9000 and 10000) and recomputed the dimensionality slopes. The results were nearly identical to those obtained using the full scale range (Figure 18A), indicating that our conclusions are not driven by edge effects.

We further repeated the analysis using a reduced sample size (1000 samples) to test the sensitivity of the results to data availability. As shown in Figure 18B, the relationship between dimensionality slope and alignment performance remained stable, demonstrating that our findings generalize across sample sizes.

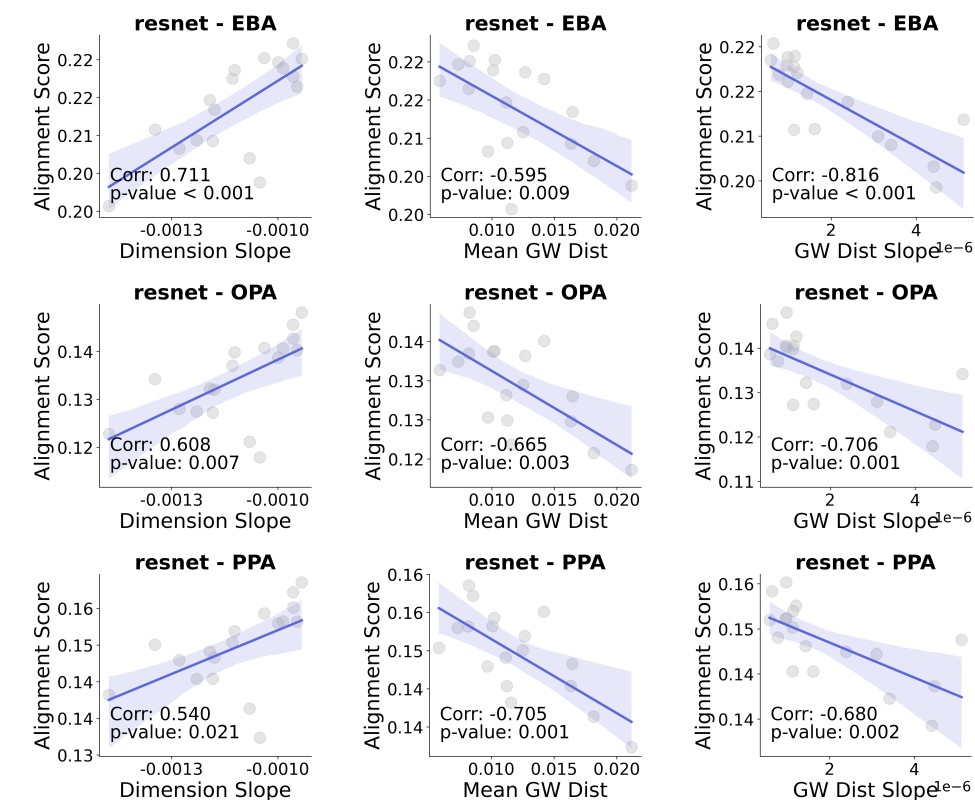

Figure 17: **Validating the relationship between alignment performance and scale-invariance metrics across ResNet architectures.** We evaluate alignment in EBA, OPA, and PPA, and find that scale-invariance metrics exhibit consistent and significant correlations with alignment performance across all ResNet variants.

Finally, we evaluated the consistency of our results across different dimensionality estimators. Using both the MOM estimator (Elmoznino & Bonner, 2024) and the MADA estimator (Farahmand et al., 2007), we again observed highly similar trends (Figure 18C and D). Together, these analyses confirm that the observed scale-invariance effects are robust across scale boundaries, sample sizes, and estimation methods.

# K    SCALE INVARIANCE ANALYSIS OF IMAGE

To test whether scale-invariance in the raw images contributes to the observed alignment, we divided the stimulus set into 20 non-overlapping subsets and computed embeddings for each subset using two models: *convnext_base.clip_laion2b_augreg* and *convnext_base.fb_in22k*. For each subset, we measured both the alignment score and the scale-invariance of the raw images. In Figure 19, we found no significant correlation between these two quantities, indicating that the alignment performance is driven by the geometric properties of the model embeddings, rather than the scale-invariance inherent in the images themselves.

# L    POWER LAWS AND THEIR RELATION TO SCALE-INVARIANCE

## L.1    POWER-LAW STRUCTURE IN NEURAL DATA

Power-law statistics frequently appear in neural activity. Covariance spectra often exhibit heavy-tailed decay of the form $\lambda_k \propto k^{-\alpha}$, indicating that information is distributed across multiple scales: a few dimensions capture large variance, while many weaker components contribute fine-grained

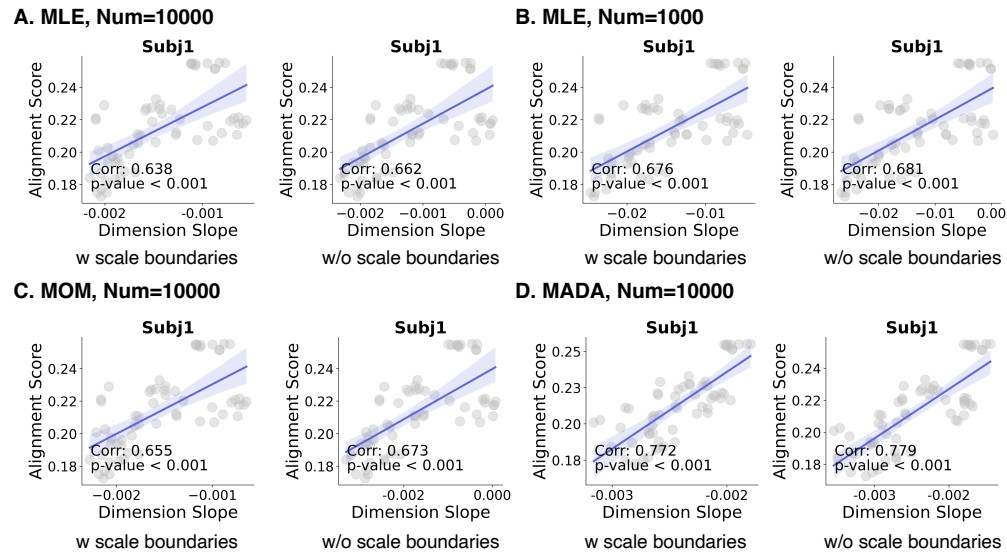

Figure 18: **Additional validation of the dimensionality analysis.** **(A)** Re-estimating dimensionality after removing boundary scales (K = 9000, 10000) yields consistent results. **(B)** The analysis remains stable when repeated with a reduced sample size (1000 samples). **(C−D)** Using alternative dimensionality estimators (MOM and MADA) produces similar trends, confirming the robustness of the dimensionalityalignment relationship.

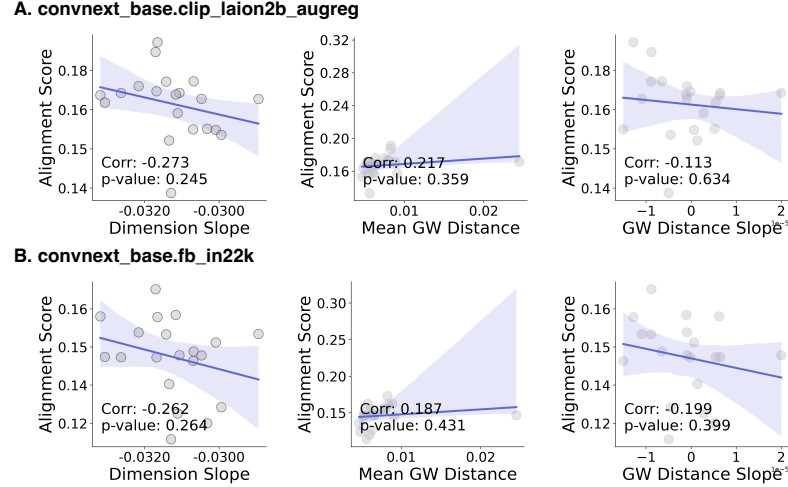

Figure 19: **Scale-invariance analysis of raw images.** Both models show no significant correlation between the scale-invariance of raw images and alignment performance.

structure. We refer to this as *information-level scale-invariance*, meaning that no characteristic variance scale dominates the representation.

However, such statistical scale-invariance alone does not guarantee alignment between datasets. For example, EEG power spectral densities (PSDs) from different subjects often share heavy-tailed profiles, yet their precise distributions vary substantially. As a result, a decoder trained on one subject may fail to generalize to another, despite both exhibiting similar power-law behavior. This demonstrates that power-law statistics constrain only the *amount* of information at each scale, not the underlying relational structure.

In contrast, *geometric* scale-invariance imposes a much stronger requirement: local neighborhoods must preserve self-similar relational structure across scales. When two neural representations share such geometric organization, their multi-scale structure becomes more compatible, making it easier for a single transformation to map one representation onto the other.

## L.2    EMPIRICAL EXAMINATION OF POWER-LAW FITS

To evaluate whether information-level scale-invariance is related to alignment, we analyzed the covariance eigenvalue spectra of embeddings from 60 pretrained vision models. For each model, we fitted a power-law model to the eigenvalue distribution and extracted two key quantities: (i) the estimated power-law exponent and (ii) the goodness-of-fit ($R^2$) of the log–log linear regression, which reflects how closely the spectrum follows a power-law decay.

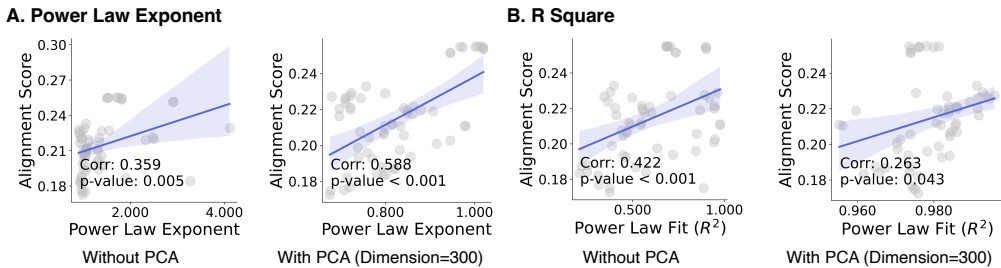

Figure 20: **Relationship between power-law statistics and alignment performance across models. (A)** Correlation between alignment scores and the power-law exponent of the covariance eigenvalue spectrum. **(B)** Correlation between alignment scores and the power-law goodness-of-fit ($R^2$).

We found that the power-law exponent exhibits a moderate correlation with alignment ($r = 0.588$) after controlling for dimensionality via PCA (Figure 20A), whereas the $R^2$ of the loglog regression shows a weaker association ($r = 0.263$) (Figure 20B). In contrast, the geometric scale-invariance metrics proposed in the main text achieve substantially higher and more robust correlations across all subjects and models.

These results indicate that while power-law statistics capture meaningful global regularities in the variance structure of embeddings, they provide limited insight into the geometric properties that more directly govern modelbrain correspondence.

## L.3    LIMITATIONS OF POWER-LAW METRICS

We believe several factors may explain why power-lawbased metrics correlate with alignment performance less strongly than the geometric measures introduced in this work:

- **Exponents do not quantify scale-invariance strength.** The exponent only reflects the decay rate of the covariance eigenvalues and does not indicate how close the embedding is to exhibiting scale-invariance.
- **$R^2$ is sensitive to outliers.** A few deviating eigenvalues can strongly influence the loglog linear fit, resulting in variability that does not reflect meaningful representational differences.
- **Fits capture only a single global scale.** Power-law regressions do not indicate whether the spectrum maintains a scale-free pattern across a wide or narrow range of components.
- **Spectral statistics ignore geometry.** Two embeddings may have nearly identical eigenvalue spectra yet possess markedly different pairwise structures and, consequently, very different alignment behavior.

## L.4    RELATION TO GEOMETRIC SCALE-INVARIANCE

The geometric scale-invariance metrics proposed in this work address precisely the types of structure that power-law statistics miss. Multi-scale intrinsic dimension characterizes how local complexity

evolves with neighborhood size, and multi-scale distributional similarity measures the stability of relational geometry across scales. Both quantities are inherently geometric and operate directly on the spatial organization of the embedding.

While power-law covariance spectra reflect scale-free patterns of information allocation, geometric scale-invariance concerns the preservation of relational structure across scales. These notions are related but distinct: the former captures global spectral regularities, whereas the latter quantifies multi-scale structural stability. Our empirical results show that geometric scale-invariance is substantially more predictive of neural alignment.

## M  ANALYSIS OF TRIVIAL FRACTAL

To assess whether our scale-invariance metrics can distinguish meaningful self-similarity from trivial randomness, we applied the same multi-scale analysis to Gaussian and uniform distributions. These distributions are sometimes described as trivial fractals: they exhibit statistical similarity across scales, but this self-similarity carries no semantic or functional structure.

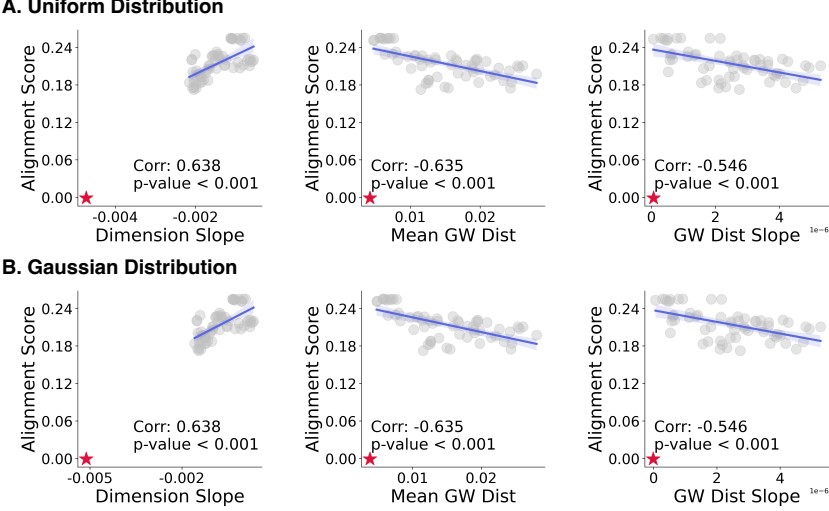

Figure 21: **Scale-invariance analysis of random distributions.** (**A**) Multi-scale intrinsic dimensionality and multi-scale GromovWasserstein distances for a uniform distribution. (**B**) Corresponding results for a Gaussian distribution. In both cases, the multi-scale GW distances are extremely small, reflecting strong distributional similarity across scales. However, the intrinsic dimensionality varies substantially with scale.

Across both Gaussian and uniform datasets in Figure 21, we observed that the multi-scale GromovWasserstein distances remained near zero at all scales. This behavior is expected: random point clouds preserve the same exchangeable geometry when rescaled, so GW distances alone cannot differentiate trivial from meaningful self-similarity.

In contrast, the slopes of local dimensionality across scales were substantially larger for Gaussian and uniform distributions than for model embeddings. This difference reflects the properties of the data: random distributions have much higher intrinsic dimensionality than learned representations, so small changes in scale produce larger variations in local dimensionality. This also indicates that when the slope is close to zero, the underlying structure must be non-trivial rather than random.

## N  WHOLE-BRAIN DISTRIBUTION OF SCALE INVARIANCE

To assess how scale-invariant structure is distributed across the cortex, we extended our analysis to all brain voxels and evaluated scale-invariance from two complementary geometric perspectives. Together, these analyses provide a whole-brain view of how consistently neural representations maintain stable structure across scales.

**A. Whole-brain Distribution of Dimension Slope**

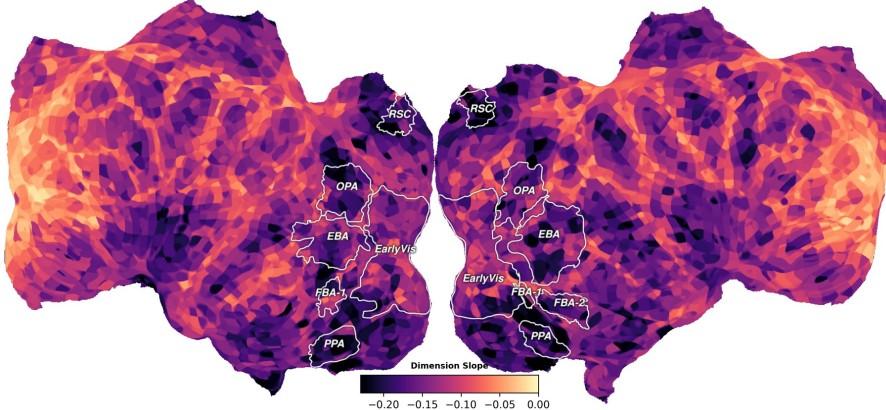

**B. Whole-brain Distribution of Gromov Wasserstein Distance Mean**

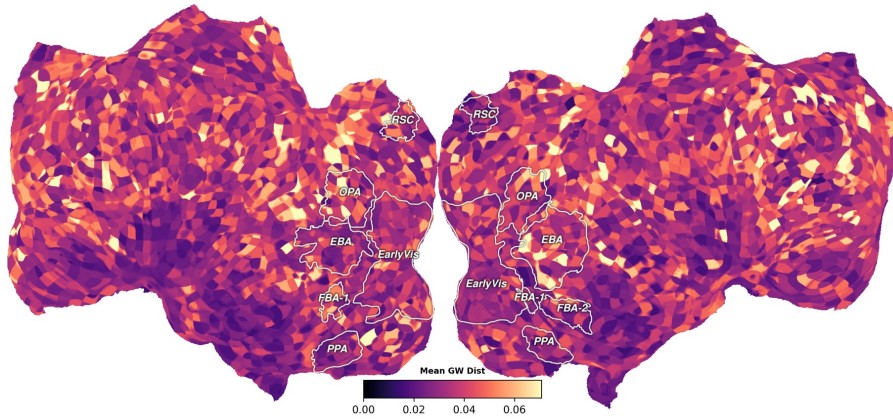

**C. Whole-brain Distribution of Gromov Wasserstein Distance Slope**

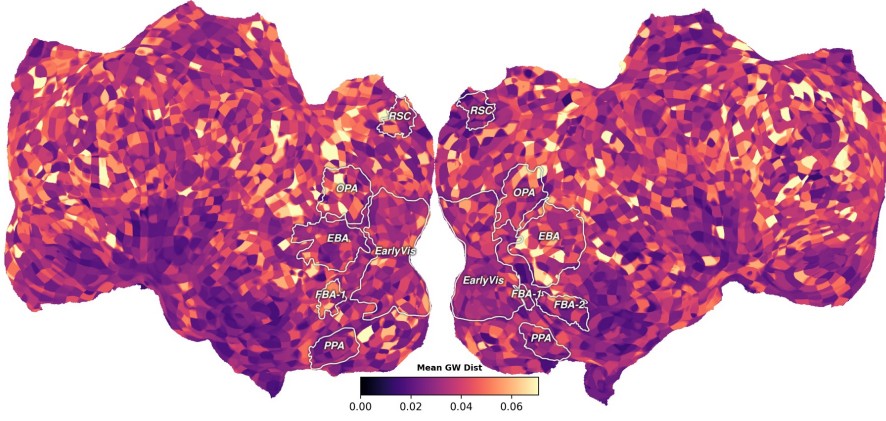

Figure 22: **Whole-brain scale-invariance analysis.** **(A)** Distribution of dimensionality slopes across all voxels. **(B)** Mean multi-scale GW distance across voxels. **(C)** Distribution of GW-distance slopes across the cortex.

**Multi-scale dimensionality and GW geometry.** We first quantified scale-invariance using the slope of local dimensionality across scales and the multi-scale Gromov–Wasserstein (GW) distance. As shown in Figure 22, most cortical voxels exhibited small dimensionality slopes, indicating stable

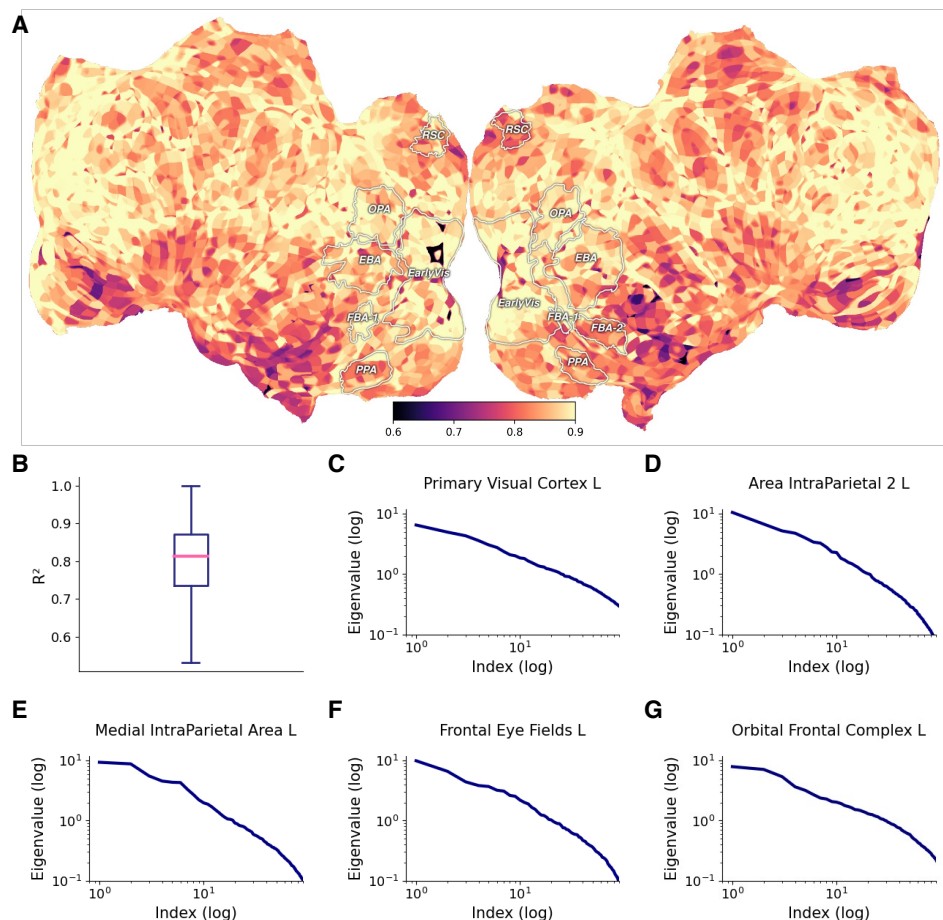

Figure 23: **Power-law analysis of local covariance spectra across the brain.** **(A)** Whole-brain distribution of voxel-wise $R^2$ values. **(B)** Histogram of $R^2$ across all voxels, showing consistently high goodness of fit. **(C–G)** Example eigenvalue spectra from cortical regions, illustrating slower decay in lower-$R^2$ areas.

intrinsic dimensionality across scales. Although some regions, notably portions of prefrontal cortex, showed slightly more pronounced stability, the overall distribution was broadly homogeneous. A similar pattern appeared in GW distances: the preservation of relational geometry across scales was consistently high throughout the brain, with minimal regional variability. Thus, both dimensionality-based and geometry-based metrics converge in showing that large-scale cortical representations exhibit strong and widespread scale-invariant organization.

**Power-law structure of local covariance spectra.** As a second and traditional measure of scale-invariance, we examined whether the eigenvalue spectra of local activity patches ($4\times4\times4$ voxels) follow a power-law decay. For each patch, we computed the eigenvalues of its covariance matrix, fitted a linear model in log–log space, and quantified goodness of fit using the $R^2$ value. The whole-brain distribution of $R^2$ values (Figure 23A and B) was uniformly high, indicating that local covariance spectra across most cortical regions closely adhere to a power-law form. To illustrate the source of regional variability, Figure 23C–G shows eigenvalues' spectra from several areas. Regions with lower $R^2$ typically exhibited more slowly decaying eigenvalue spectra, leading to heavier tails.

Across both analyses, the cortex shows strong and widely distributed scale-invariance. Small deviations are primarily driven by slower spectral decay in certain regions rather than a breakdown of multi-scale organization. These consistent patterns demonstrate that scale-invariance is a global property of neural representations across the human cortex.

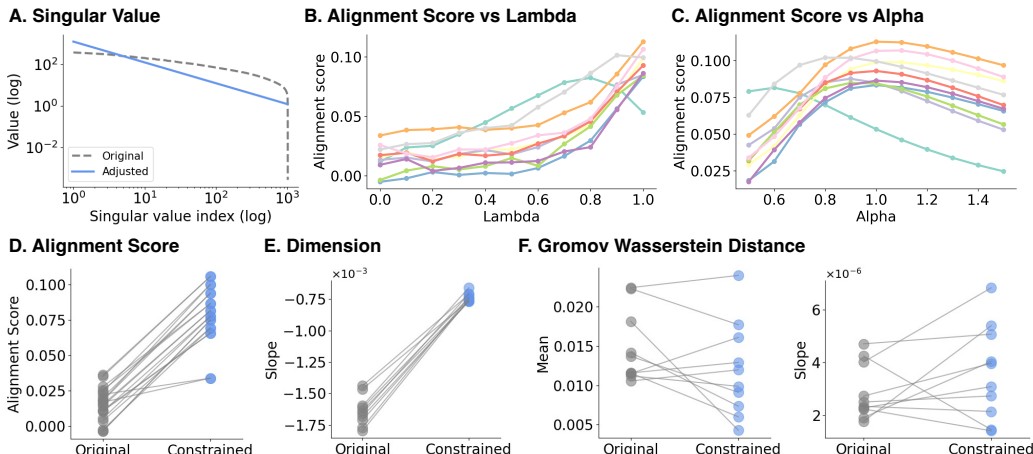

Figure 24: **Manipulating embedding singular values improves alignment.** **(A)** Illustration of the singular-value manipulation procedure, where the original spectrum is replaced by a convex combination with a target power-law spectrum. **(B)** Effect of manipulation strength $\lambda$ on alignment performance. **(C)** Effect of changing the power-law exponent $\alpha$ (with $\lambda = 1$) on alignment. **(D)** Alignment improvement before and after manipulation for the 10 lowest-performing models. **(E)** Changes in dimensionality slopes before and after manipulation. **(F)** Changes in the slope and mean of multi-scale Gromov–Wasserstein distances before and after manipulation.

## O    MANIPULATING EMBEDDING SINGULAR VALUES IMPROVES ALIGNMENT

Our goal in this experiment is to directly adjust the scale-invariance of embeddings and examine whether alignment performance changes accordingly. To do so, we manipulated the singular value distributions of the embedding matrices. Specifically, for each model we computed its singular values and replaced them with a convex combination of the original distribution and a prescribed power-law distribution with exponent $\alpha$. The parameter $\lambda \in [0, 1]$ controls the manipulation strength: $\lambda = 0$ leaves the embedding unchanged, while $\lambda = 1$ fully replaces its singular values with the target power law. This procedure enforces a precise power-law spectrum (Figure 24A), allowing us to examine how controlled changes in information-level scale-invariance affect alignment.

We focus on the 10 models with the lowest alignment performance. To isolate the effect of singular-value manipulation, we fixed the ridge-regression regularization parameter to $\lambda_{\text{ridge}} = 1$ for all evaluations. Figure 24B shows the effect of varying the manipulation strength under a fixed exponent $\alpha = 1$: alignment improves monotonically as $\lambda$ increases, indicating that enforcing a cleaner power-law structure in the singular values consistently enhances alignment.

Next, fixing $\lambda = 1$, we varied the exponent $\alpha$ (Figure 24C). Alignment increases as $\alpha$ grows from small values, peaks around $\alpha \approx 1$, and then declines. The optimal $\alpha$ differs slightly across models, demonstrating that the exponent is not monotonically related to alignment and cannot be used as a standalone indicator of scale-invariance quality.

Figure 24D summarizes alignment improvements before and after manipulation for the 10 lowest-performing models under $\alpha = 1$ and $\lambda = 1$, showing a consistent and substantial boost. To understand which aspects of scale-invariance are affected, we examined changes in our multi-scale metrics. Dimensionality slopes decrease markedly after manipulation (Figure 24E), indicating more stable multi-scale dimensional structure. In contrast, the slope of multi-scale GromovWasserstein distances shows no clear systematic change, and its mean shifts only slightly (Figure 24F).

These results highlight two key points. First, manipulating singular values primarily alters the distribution of information across components, improving multi-scale dimensional stability but only weakly influencing the geometric consistency captured by GW distances. This aligns with earlier findings: power-law behavior in the singular value or covariance spectrum correlates with alignment, but far less strongly than geometric scale-invariance. Second, our manipulation operates *after* embeddings are computed and cannot induce geometric scale-invariance in the representation itself.

This suggests that future work should use geometric scale-invariance metrics to guide representation learning directly, rather than modifying spectrum statistics post hoc.

## P  LLM USAGE STATEMENT

In accordance with the ICLR 2026 policy on responsible usage of Large Language Models (LLMs), we disclose that LLMs were employed to aid in the preparation of this manuscript. Specifically, LLMs were used to polish writing, improve clarity and refine grammar. All ideas, analyses and conclusions presented in this work are solely those of the authors.

