# OpenReview forum: "Scale-Invariance in AI Representation Predicts AI-Brain Alignment"
_ICLR.cc/2026/Conference — Submitted to ICLR 2026_

### Official Review · Reviewer_Eohs · 2025-10-30

**Soundness:** 2
**Presentation:** 3
**Contribution:** 2
**Rating:** 2
**Confidence:** 4

**Summary:**

This study explored how a specific factor, namely, scale-invariance, is correlated to the alignment between visual AI models and brain responses at embedding-level. The authors propose two metrics, multi-scale dimensional stability and multi-scale distributional similarity, to measure scale-invariance, and build linear regression model to assess the predictive relationship between scale-invariance and model-brain alignment score. The experimental results demonstrate that both metrics predict neural alignment.

**Strengths:**

1.The authors have proposed two metrics to measure the scale-invariant property of embeddings in visual AI models.
2.The authors have recruited 60 visual models to establish the predictive relationship between scale-invariant and model-brain alignment.
3.The authors have demonstrated that larger pre-training datasets enhance scale invariance and consequently improve alignment, whereas fine-tuning reduces scale invariance, leading to weaker alignment.

**Weaknesses:**

1.Computational and neuro-scientific justifications of the results are very limited.
2.Justifications about model selection are missing.
The definition of “scale-invariance” is different from common understanding, potentially miss leading.

**Questions:**

1.The author should provide interpretation of, at least discussion about, the AI models’ scale-invariance. Are the AI modes’ scale-invariance measured by the two metrics intuitively or quantitatively align with their architectures/characteristics? This point is quite important for the readers to assess the feasibility of the proposed metrics.
2.The predictive relationship between model-brain alignment and scale-invariance looks quite similar across brain regions. The question is: which brain regions encode scale-invariance predominantly, or scale-invariance is encoded in widely distributed brain regions? Is there any evidence in neuroscience or cognitive neuroscience that supports your results?
3.Are their any reasons for AI model selection? Are those models theoretically engage scale-invariance in a wide range? The slops in Figure 4 and Figure 5 are very small. Does it mean the difference of scale-invariance across model is small? Meanwhile, such small slops may make the regression unstable, challenging the effectiveness of the significance level.
4.The definition of scale-invariance in visual perception is somehow different from common understanding. Scale-invariance, in most cases, means that certain perceptual judgments, neural representations, or visual computations remain essentially the same at different physical sizes or viewing distances.
5.How can the findings in this study help to develop brain-like AI? A short discussion may help the reader to understand the study.

---

> ### Author Response · Authors · 2025-11-25
>
> > Q1. Computational and neuro-scientific justifications of the results are very limited.
>
> Extensive neuroscience research has demonstrated that neural activity exhibits clear scale-invariant characteristics. Classic evidence includes the observation that either the eigenvalue spectrum of the covariance matrix or the variance explained by principal components often follows a power-law distribution [1, 2]. Inspired by this, we hypothesize that the brain organizes its representations in a manner that preserves scale-invariant structure. Consequently, AI models that better align with brain activity are expected to exhibit embeddings that reflect similar geometric properties, including cross-scale consistency and self-similarity.
>
> Based on this reasoning, we introduced a geometric scale-invariance framework combining multi-scale dimensional stability and multi-scale distribution similarity. This framework quantitatively characterizes whether embeddings maintain consistent structure across scales. Our experiments show that models with higher alignment scores possess embeddings closer to geometric scale-invariance, and that models pretrained on larger datasets exhibit stronger self-similarity. This suggests that AI models not only approximate brain functionally but also reflect similar organizational principles at the representational level, which we consider a core contribution of this work.
>
> > Q2. Justifications about model selection are missing. The definition of “scale-invariance” is different from common understanding, potentially miss leading.
>
> We selected the ConvNeXt family primarily for experimental controllability rather than any inherent theoretical predisposition toward scale-invariance. Within this family, models vary systematically in pretraining dataset size (ImageNet-1K, ImageNet-12K, ImageNet-22K, LAION-A, LAION-2B), pretraining paradigm (standard image classification vs. multimodal CLIP-style pretraining), and parameter count, while architectural design details remain largely consistent. This design enables us to isolate the effect of data and pretraining while minimizing confounding factors.
>
> To ensure robustness across architectures, we also analyzed ResNet models. As shown in **Appendix I, Figure 17**, the scale-invariance metrics continue to reliably reflect differences in alignment, confirming cross-architecture consistency.
>
> Regarding the definition of scale-invariance, we adopt a mathematically general concept and translate it into a geometric interpretation: “fractal-like structure as cross-scale consistency.” In neuroscience, “scale-invariant recognition” often refers to perceptual invariance to object size, whereas our focus is on the stability of the geometric structure of embeddings across scales. These are distinct perspectives: we are not assessing perceptual invariance, but rather the geometric self-consistency of embeddings at the representation level.
>
> > Q3.Interpretation of AI models’ scale-invariance.
>
> The scale-invariance we measure is a property of the model embeddings, not of the network architecture itself. Specifically, we examine whether the geometric structure of embeddings, including dimensionality stability and distributional consistency, remains stable when local neighborhoods expand to larger scales. In other words, scale-invariance is an embedding-level, data-driven property rather than a pre-defined architectural feature.
>
> Systematic experiments indicate that no particular architecture inherently exhibits stronger scale-invariance, instead, data factors dominate. For instance, **Figure 6 (Section 4.4)** shows that increasing pretraining dataset size within the same architecture family significantly enhances embedding scale-invariance. Moreover, improvements in scale-invariance occur in parallel with improvements in alignment, demonstrating that scale-invariance reflects a data-driven geometric property of the representations. This is consistent with previous findings that model architecture has limited impact on alignment, whereas dataset size plays a more significant role [3].
>
> We also validated this on ResNet architectures (**Appendix I**), where the results show that the scale-invariance metrics continue to reliably reflect differences in alignment. The feasibility of these metrics stems not only from robust experimental results but also from well-established theoretical foundations: dimensional stability relies on mature MLE-based intrinsic dimension estimation algorithms, and distribution similarity is grounded in optimal transport theory. Together, these metrics provide a quantitative and interpretable measure linking embedding geometry to neural alignment.

---

> > ### Author Response · Authors · 2025-11-25
> >
> > > Q4.Brain region specificity of scale-invariance.
> >
> > The similar shape of alignment–metric curves across visual regions is expected. In vision-related areas, models that perform well in one region typically perform well in others, resulting in stable relative rankings. Hence, cross-region plots naturally appear similar, reflecting model-level consistency rather than lack of regional specificity.
> >
> > We assessed the whole brain (**Appendix N, Figure 22**) and found no regions with scale-invariance metrics consistently higher than others. This suggests that scale-invariance is not dominated by any specific brain region. Crucially, alignment depends more on the information encoded by a region than on its intrinsic scale-invariance. Our central finding remains: embeddings with stronger scale-invariance achieve higher alignment in vision-related areas.
> >
> >
> > > Q5.Are their any reasons for AI model selection? Are those models theoretically engage scale-invariance in a wide range? The slops in Figure 4 and Figure 5 are very small. Does it mean the difference of scale-invariance across model is small? Meanwhile, such small slops may make the regression unstable, challenging the effectiveness of the significance level.
> >
> > ConvNeXt models were chosen for their controllable pretraining variables, not because they are theoretically predisposed to scale-invariance. Dataset size, pretraining paradigm, and model scale vary systematically while architecture remains consistent, enabling clean analysis of their effects. Key experiments were repeated on ResNet models (**Appendix I, Figure 17**), showing consistent results, confirming robustness across architectures.
> >
> > Importantly, our findings show that scale-invariance is not an inherent architectural property but is learned from training data. As dataset size increases, embeddings exhibit stronger scale-invariance (**Figure 6**), supporting a data-driven mechanism. This is consistent with previous work suggesting that dataset size, rather than model architecture, primarily drives alignment [3].
> >
> > Regarding small slopes in Figures 4 and 5, this reflects the scale of the axes: the horizontal scale spans three orders of magnitude, whereas vertical dimension changes are on the order of single digits. The small numerical slope does not imply insignificant differences or regression instability. Normalizing the scale would yield larger slope values, but statistical significance remains unaffected. The linear relationships are clear and robust.
> >
> > > Q6.The definition of scale-invariance in visual perception is somehow different from common understanding.
> >
> > In perception literature, scale-invariance typically refers to consistent recognition across changes in object scale. Our work considers geometric scale-invariance in embedding space, focusing on the stability of high-dimensional structures across scales. The proposed metrics, multi-scale distribution similarity and multi-scale dimensional stability, assess embeddings’ geometric consistency at multiple scales. While this perspective differs from perceptual scale-invariance, it is grounded in well-established mathematical theory, including fractal geometry and optimal transport, providing a rigorous and quantitative framework for analysis [4, 5].

---

> > > ### Author Response · Authors · 2025-11-25
> > >
> > > > Q7.How can the findings in this study help to develop brain-like AI? A short discussion may help the reader to understand the study.
> > >
> > > Our findings provide a clear, actionable pathway for developing brain-like AI. By using multi-scale geometric metrics, the structure of model embeddings can be evaluated without relying on neural data, making brain-likeness a diagnosable and quantifiable property during model design and training.
> > >
> > > We show that alignment with the brain is strongly determined by the multi-scale organization of embeddings. Because brain-likeness reflects this geometric property rather than a post-hoc fit to neural data, models can be trained to enhance multi-scale consistency and reduce structural distortions, gradually shaping embeddings toward a brain-like organization.
> > >
> > > Importantly, this approach does not require extensive neural recordings, is scalable, and can serve as a general inductive bias in representation learning. Overall, our work demonstrates that brain-likeness can be quantified, predicted and directly guided through embedding geometry, providing a biologically interpretable framework for AI design.
> > >
> > > **Reference**
> > >
> > > [1] Gauthaman, Raj Magesh, Brice Ménard, and Michael F. Bonner. "Universal scale-free representations in human visual cortex." arXiv preprint arXiv:2409.06843 (2024).
> > >
> > > [2] Stringer, Carsen, et al. "High-dimensional geometry of population responses in visual cortex." Nature 571.7765 (2019): 361-365.
> > >
> > > [3] Conwell, Colin, et al. "A large-scale examination of inductive biases shaping high-level visual representation in brains and machines." Nature communications 15.1 (2024): 9383.
> > >
> > > [4] Mandelbrot, Benoit B. "Fractal geometry: what is it, and what does it do?." Proceedings of the Royal Society of London. A. Mathematical and Physical Sciences 423.1864 (1989): 3-16.
> > >
> > > [5] Falconer, Kenneth. Fractal geometry: mathematical foundations and applications. John Wiley & Sons, 2013.

---

> > ### Comment · Reviewer_Eohs · 2025-11-27
> >
> > I appreciate the authors’ detailed responses. Several concerns remain.
> > 1) About neuro-scientific justifications: As the authors claimed, “scale-invariance is not dominated by any specific brain region”, and model-brain alignment is improved across nearly all sub-regions in the visual cortex. The question is, what are the mainstreams in neuroscience regarding to neural encoding of “scale-invariance”? Are your claims and observations in agreement or conflicting with prior neuroscience evidences? If your observations conflict with prior neuroscience findings, how to interpret your results? This is quite important in my opinion since it may fundamentally overturn your major findings.
> >
> > 2) What exactly are the range of the two metrics, either within-model or across-model?

---

> > > ### Author Response · Authors · 2025-12-01
> > >
> > > We sincerely thank the reviewer for the constructive follow-up questions. We address each point below.
> > >
> > > **1. Neuro-scientific justifications of scale-invariance**
> > >
> > > In neuroscience, scale-invariance is a well-established concept and is understood as a **global property of the brain**, not something confined to any specific region. Prior literature typically categorizes scale-invariance into three major forms:
> > >
> > > (1) **Structural** (e.g., cortical folding patterns, dendritic fractal morphology),
> > >
> > > (2) **Topological** (e.g., scale-free characteristics in structural brain networks), and
> > >
> > > (3) **Dynamical** (e.g., neuronal avalanches, power-law frequency spectra, long-range temporal correlations) [1–4].
> > >
> > > These forms of scale-invariance reflect widespread, brain-wide organizational principles rather than region-specific phenomena [5], and they play important functional roles in neural computation and information processing [6].
> > >
> > > Our findings are **fully consistent** with existing neuroscience evidence. We additionally conducted a whole-brain analysis of the fMRI covariance eigenvalue spectra (**Appendix N, Fig. 23**). Using a linear fit in log–log space and quantifying power-law conformity with the resulting $R^2$, we observed consistently high $R^2$ values across the whole brain (Fig. 23B, **mean=0.75**). This again confirms that scale-invariance is a global feature of neural dynamics rather than a property dominated by individual regions.
> > >
> > > **2. Range of the proposed metrics**
> > >
> > > Our proposed metrics (Multi-scale dimensional stability and Multi-scale distributional similarity) are computed from the embeddings produced by each model on the same dataset. Therefore, each model corresponds to exactly one value per metric, and there is no notion of “within-model range,” unlike in the brain, where regional differences can be computed.
> > >
> > > The concept of “range” is thus meaningful only across models. For the set of models evaluated in our experiments, the across-model variation of the key quantities is approximately:
> > >
> > > * Dimension Slope: (−0.002, −0.001)
> > > * Mean GW distance: (0.005, 0.025)
> > > * GW distance Slope: (0, 5e-6)
> > >
> > > These ranges reflect the relative differences in the multi-scale geometric properties of embeddings across different models.
> > >
> > >
> > >
> > > **Reference**
> > >
> > > [1] Roberts, James A., Tjeerd W. Boonstra, and Michael Breakspear. "The heavy tail of the human brain." Current opinion in neurobiology 31 (2015): 164-172.
> > >
> > > [2] Khaluf, Yara, et al. "Scale invariance in natural and artificial collective systems: a review." Journal of the royal society interface 14.136 (2017): 20170662.
> > >
> > > [3] Stringer, Carsen, et al. "High-dimensional geometry of population responses in visual cortex." Nature 571.7765 (2019): 361-365.
> > >
> > > [4] O’Byrne, Jordan, and Karim Jerbi. "How critical is brain criticality?." Trends in Neurosciences 45.11 (2022): 820-837.
> > >
> > > [5] Wang, Zezhen, et al. "The geometry and dimensionality of brain-wide activity." eLife 14 (2025): RP100666.
> > >
> > > [6] Kardan, Omid, et al. "Improvements in task performance after practice are associated with scale-free dynamics of brain activity." Network Neuroscience 7.3 (2023): 1129-1152.

---

### Official Review · Reviewer_UeYU · 2025-10-31

**Soundness:** 2
**Presentation:** 3
**Contribution:** 2
**Rating:** 2
**Confidence:** 3

**Summary:**

The paper claims that “scale invariance” of embeddings explains when model features align with fMRI, measured via multi-scale dimensional stability and a Gromov–Wasserstein multi-scale distributional similarity. On NSD with 60 vision models, stronger scale invariance predicts higher voxelwise R^2. Larger pretraining datasets raise it and alignment, while fine-tuning lowers both. The authors argue that these metrics add explanatory power beyond model size and could be used as regularizers for making more brain-like geometry.

**Strengths:**

- Novel central hypothesis that raises an interesting point. I would say their work is original. If my concerns can be addressed, this work has good significance.
- Great care is taken care into the analysis (except for the points raised in the weaknesses section below).
- Clearly written.

**Weaknesses:**

I am mainly confused as to why the authors resorted to quantifying the scale invariance with "Multi-scale dimensional stability" and "Multi-scale distributional similarity." I do not believe they are a good measure of scale-invariance, since they cannot distinguish between trivial scale-invariance and more interesting scale-invariance. I am not quite convinced with the author's claim "While effective for continuous data, these methods often struggle with discrete datasets and provide limited ways to quantify the degree of scale-invariance." in line 120.

In neuroscience, scale-invariance is typically characterized by the power-law statistics, as noted by the authors (the authors should also cite the landmark paper [1] on this topic). By power-law statistics, I am specifically referring to the power law in the eigenvalue spectrum of the covariance matrix of neural representations. The power-law characteristic is directly linked to a non-trivial fractal structure.

The problem with the slope analyses on "Multi-scale dimensional stability" and "Multi-scale distributional similarity" is that they will indicate that a simple uniform distribution is scale-invariant. It is technically true that a uniform distribution is scale-invariant, but that is like saying a straight line is a fractal curve (which is true, but not interesting). When we discuss scale-invariance in neuroscience, we are not referring to trivial scale-invariance like that. I think the authors are aware of that, as they use the conventional nontrivial fractal cartoon in Figure 1.

My intuition is that a Gaussian distribution would also result in small slopes in both the "Multi-scale dimensional stability" and "Multi-scale distributional similarity" analyses (but I could be wrong).

The paper currently requires a stronger justification of the claim in line 120. Also, I believe the paper's claim can be strengthened if they simply report the eigenvalue spectra of neural recording, fit the power-law curve to it, and report perhaps the fitted exponent against the alignment score and/or the slope score.

I should reiterate that I find the hypothesis of this paper very intriguing, and it is something very much worth investigating. I just simply skeptical of the author's methodology.

[1] Stringer et al. 2019 "High-dimensional geometry of population responses in visual cortex"

**Questions:**

Typo in line 264: “perdormance”

---

> ### Author Response · Authors · 2025-11-25
>
> > Q1.Distinguishing trivial and non-trivial scale-invariance.
>
> We thank the reviewer for the suggestion. We added a control experiment (**Appendix M**) to examine the behavior of uniform and Gaussian distributions under our two metrics. The results show that multi-scale Gromov–Wasserstein (GW) distances are indeed small for random distributions, because random point clouds are statistically similar across scales. However, the dimensionality changes dramatically with scale, and the slope of dimensionality is large.
>
> This observation motivates the use of our two complementary metrics. Considering only multi-scale GW distance can easily misclassify random data as “scale-invariant,” but incorporating dimensionality stability clearly excludes trivial distributions. Together, these metrics effectively distinguish between trivial “random self-similarity” and true structured scale-invariance.
>
> Moreover, in high-dimensional embeddings, differences in dimensionality across scales are naturally amplified. Therefore, if we observe that dimensionality remains relatively stable as the scale changes, this provides strong evidence that the embedding forms a stable, non-trivial structure, further distinguishing it from trivial fractal patterns.
>
> Finally, neural network embeddings are typically low-dimensional rather than random, trivial distributions. Therefore, when analyzing embeddings, there is little risk of misclassifying trivial structures as non-trivial scale-invariance.
>
> > Q2. Power-law characteristics and discrete data (line 120).
>
> We thank the reviewer for the comment. We have revised the text to clarify that the power-law property of discrete data can indeed be analyzed via the eigenvalues of the covariance matrix.
>
> Mathematically, the power-law exponent only describes the decay rate of the spectrum and does not fully capture the multi-scale geometric structure. Only the linearity of the eigenvalue distribution in log–log space truly reflects the degree of scale-invariance. While fitting a linear model and computing R^2 provides a quantitative measure, this method is sensitive to outliers, boundary effects and the choice of fitting interval, and thus is not a perfect metric.
>
> Moreover, although some data (e.g., EEG power spectral density) exhibit power-law spectra, this does not guarantee high alignment between datasets. In fact, EEG signals show substantial variability both within and across subjects, demonstrating that satisfying a power-law does not directly lead to alignment. Therefore, when studying AI–brain alignment, we argue that the scale-invariance of the data’s geometric structure should be considered. This motivates the geometric scale-invariance metrics proposed in our work.
>
> > Q3.report the eigenvalue spectra of neural recording, fit the power-law curve to it, and report perhaps the fitted exponent against the alignment score and/or the slope score.
>
> We followed the reviewer’s suggestion and included an analysis of covariance spectra in **Appendix L**, reporting both the fitted power-law exponent and the log–log linear fit R^2. These metrics are significantly correlated with alignment scores, but their correaltion are weaker than our proposed scale-invariance metrics. This indicates that while power-law spectra capture some statistics relevant to scale-invariance and alignment, they do not fully reflect the multi-scale geometric structure.
>
> From a methodological perspective, the R^2 from log–log fitting is highly sensitive to noise and fitting intervals, making it difficult to directly optimize embeddings for scale-invariance. In contrast, our multi-scale GW distance and related geometric metrics are more amenable to optimization, allowing scale-invariance to be both evaluated and potentially enhanced during model training.
>
> We also added a key manipulation experiment in **Appendix O**: by directly adjusting the singular values of embeddings to strictly follow a given power-law exponent, we observed that this spectral adjustment improves alignment and increases the stability of multi-scale dimensionality. However, the multi-scale GW distance remains unchanged. This demonstrates that:
>
> (1)Adjusting the singular value spectrum can improve statistical scale-invariance and partially enhance alignment.
>
> (2)But modifying the power-law spectrum alone does not alter the geometric structure, the GW-based geometric scale-invariance remains unchanged.
>
> These observations suggest that improving alignment requires not only optimizing the singular value distribution but also training models to learn representations with consistent geometric structure across scales, achieving embeddings that approach non-trivial, fractal-like scale-invariance. This is the main motivation for our geometric scale-invariance metrics.

---

### Official Review · Reviewer_XA1n · 2025-11-02

**Soundness:** 3
**Presentation:** 4
**Contribution:** 3
**Rating:** 6
**Confidence:** 3

**Summary:**

The paper studies which factors drive the alignment between neural network representations and fMRI responses on the Natural Scenes dataset by analyzing over 60 different pretrained ConvNext models. Rather than correlating individual training factors with the degree of brain alignment, they examine the embedding space directly and quantify the level of scale-invariance within the embeddings. To measure the distribution similarity between two scales, the authors employ the Gromov-Wasserstein distance between embeddings of samples at different scales, obtained by considering different neighborhood sizes from a reference point. The study finds that embeddings with stronger scale-invariance align better with fMRI responses.

**Strengths:**

* The motivation is clearly described, the methodological parts are outlined in an understandable way and the results and figures are well crafted.

* The paper studies an interesting question and the connection between scale invariance and brain alignment is novel.

* As outlined in the conclusion section, the findings and the proposed metric provides a direct, actionable approach for model design (optimizing scale invariance during pretraining) which could lead to interesting follow-up work.

**Weaknesses:**

* As far as I could see, the models in Table 1 only included image/text models and supervised models trained on ImageNet variants. Thereby the range of different pretraining objectives is limited which has been shown to be an important factor for human alignment [1]. It would be interesting to also include self-supervised models in the analysis. Further, only one architecture is included (ConvNext). While this is good for comparing other factors, an ablation should be performed showing whether this insight also holds on different model architectures.

* It would be good to show more ablations which model properties (training dataset, scale etc.) are correlated with scale invariance and how they are correlated with brain alignment. I have concerns that scale invariance might be only a proxy variable which emerges from some other property (e.g. model scale) which is also positively correlated with brain alignment and not the true causal factor.

[1] Muttenthaler, Lukas, et al. "Human alignment of neural network representations." ICLR 2023.

**Questions:**

* How is the train/hold-out split described in section 3.5 performed. Is it also across subjects?

* Did you make experiments if the scale invariance can also be estimated on a general dataset like ImageNet-1k and then used to predict the brain alignment for the Natural Scenes dataset?

---

> ### Author Response · Authors · 2025-11-25
>
> > Q1. While this is good for comparing other factors, an ablation should be performed showing whether this insight also holds on different model architectures.
>
> To ensure the generality of our findings, we supplemented our study with results on ResNet models. As shown in **Appendix I, Figure 17**, all three scale-invariance metrics remain significantly correlated with alignment scores, consistent with the main results.
>
> > Q2. It would be good to show more ablations which model properties (training dataset, scale etc.) are correlated with scale invariance and how they are correlated with brain alignment.
>
> In **Figure 6** of the main text, we examine the effect of dataset size on scale-invariance. Larger training datasets generally lead to stronger scale-invariance. However, even models pretrained on the same dataset can show different alignment scores, indicating that dataset size alone does not fully explain alignment differences. In contrast, our scale-invariance metrics can account for these differences, demonstrating that scale-invariance is not simply a proxy for dataset size.
>
> In **Appendix G**, we analyze the relationship between model parameter and scale-invariance. We find that while larger models tend to achieve higher alignment scores, increasing parameter count does not significantly enhance scale-invariance. Thus, parameter count and scale-invariance explain alignment from complementary perspectives. Multivariate regression analysis confirms that including scale-invariance metrics significantly improves the model’s R^2, demonstrating that scale-invariance is not merely a proxy for model size.
>
> Moreover, even when model size or training dataset size is unknown, scale-invariance metrics can be used to assess a model’s potential alignment with neural activity. This can also guide checkpoint selection during pretraining to optimize alignment and reduce resource waste from overtraining. These applications highlight the unique value of scale-invariance, which cannot be fully captured by dataset size or model parameters alone.
>
> > Q3. How is the train/hold-out split described in section 3.5 performed. Is it also across subjects?
>
> Analyses are performed separately for each subject. For each subject, there are 10,000 stimulus samples, of which 80% are used for training and 20% for testing.
>
> > Q4. Did you make experiments if the scale invariance can also be estimated on a general dataset like ImageNet-1k and then used to predict the brain alignment for the Natural Scenes dataset?
>
> First, the NSD dataset consists of fMRI responses collected under COCO stimuli, so we cannot directly analyze other datasets.
>
> Second, we supplemented our study with experiments to test whether the scale-invariance of the raw stimulus images affects alignment between AI models and brain activity. We divided the stimulus images into 20 subsets and computed both the alignment score and the original images’ scale-invariance for each subset. We performed this analysis on two models, convnext_base.clip_laion2b_augreg and convnext_base.fb_in22k (**Appendix K, Figure 19**). We found no significant relationship between the original images’ scale-invariance and alignment scores.
>
> We bellieve that alignment is primarily determined by how the model represents the data, rather than properties of the raw images themselves. Therefore, scale-invariance measured in the embedding space better reflects differences in alignment performance.
>
> If our interpretation of your question is incorrect, please let us know, we are happy to perform additional analyses to address any remaining concerns.

---

### Official Review · Reviewer_qqDn · 2025-11-03

**Soundness:** 3
**Presentation:** 2
**Contribution:** 2
**Rating:** 4
**Confidence:** 4

**Summary:**

The paper proposes scale-invariance as an embedding-level property that predicts AI–brain alignment. Using NSD fMRI and embeddings from 60 ConvNeXt/CLIP variants, the authors quantify scale-invariance via (i) dimensional stability and (ii) multi-scale distributional similarity (Gromov–Wasserstein across neighborhoods). Models with flatter dimensional slopes and lower/flatter GW curves align better with fMRI, especially in EBA; larger pretraining datasets increase these properties, while supervised fine-tuning reduces them (via label-centric clustering). Although model size correlates with alignment, scale-invariance adds substantial predictive power. They suggest optimizing these metrics (e.g., GW regularization) to induce more brain-like representations.

**Strengths:**

The paper presents an interesting and well-motivated idea: that scale-invariance of embeddings—quantified via dimensional stability and Gromov–Wasserstein distributional similarity—predicts alignment between visual model representations and fMRI responses. The work is clearly written and conceptually relevant to ongoing efforts to identify what factors underlie brain–model correspondence. However, several aspects limit the interpretability and impact of the findings, as outlined below.

**Weaknesses:**

1. There is limited contextualization of prior work. Section 4.1 reproduces a standard voxelwise encoding-model alignment analysis, yet omits discussion or citation of the extensive earlier literature that established this methodology (e.g., Yamins et al., 2014; Khaligh-Razavi & Kriegeskorte, 2014; Cichy et al., 2019; Conwell et al., 2024). Similarly, when stating that “recent studies show that neural networks, although never trained on neural data, often align well with brain activities,” the authors should also cite older foundational work from 2014–2017 that first demonstrated this phenomenon. Adding this historical context would help situate the contribution more accurately.

2. Model selection seems narrow. Although the paper analyzes 60 models, all are ConvNeXt variants. Since the embeddings are from pretrained checkpoints, expanding to other architectures (e.g., ResNets, ViTs, SWINs, CLIP) would require little additional effort and would make the results more general and compelling. Restricting to a single architectural family limits the interpretability of “embedding-level” conclusions.

3. My biggest concern with these and other studies is the reliance on purely correlational analysis. The study is correlational throughout—no causal manipulations are performed. Relationships between scale-invariance and alignment are inferred by comparing pretrained and fine-tuned models, not by experimentally controlling scale-invariance or dataset size. Stronger claims about causality would require training interventions, such as explicitly regularizing scale-invariance or systematically varying data scale while holding other factors fixed.

4. I also had some concerns about the interpretation of the dimensional-stability slope. In Figure 4A, all models exhibit a sharp uptick in dimensionality between the last two scales, even though alignment is not a function of neighborhood size. This pattern may reflect boundary or estimator effects rather than a meaningful geometric difference. The authors should check whether the reported correlations hold when excluding extreme scales or when using alternative intrinsic-dimensionality estimators.

5. I also think there may be possible analysis artifacts. Figure 9 shows identical correlation values (up to three decimal places) for PPA and RSC, suggesting a copy–paste or code error that should be verified.

6. Low alignment in early visual regions. Reported alignment values for V1–V4 are unexpectedly low compared to prior work showing that early visual cortex can be modeled quite accurately by CNNs and other models (St-Yves et al., 2023; Saha et al., 2025). Clarifying methodological differences—e.g., voxel inclusion criteria, PCA compression, layer choice, or regression setup—would help explain this discrepancy.

7. Unclear intuitive notion of “scale.” The definition of scale is introduced mathematically (via K-nearest neighborhoods) but remains conceptually opaque. A brief intuitive description—such as “smaller scales capture local neighborhood geometry, while larger scales reflect global embedding organization”—would help readers grasp the idea early on, ideally in the abstract or introduction.

8. Missing relevant references. The authors should also consider citing Elmoznino & Bonner (2024), which discusses scale-free structure in visual representations, and related work on covariance spectra or manifold smoothness to better position their contribution.

**Questions:**

Questions are outlined above in the weaknesses section.

---

> ### Author Response · Authors · 2025-11-25
>
> > Q1. There is limited contextualization of prior work. Authors should also cite older foundational work from 2014–2017 that first demonstrated this phenomenon.
>
> Thank you for the suggestion. We have added the relevant foundational references from 2014–2017 to both the Introduction and Related Works sections.
>
> > Q2. Model selection seems narrow.
>
> Our model set now includes the ConvNeXt family, which covers multimodal pretraining paradigms such as CLIP. This ensures a diverse set of datasets and training tasks while maintaining good control over confounding variables, which is the main reason we chose ConvNeXt for our analyses. Additionally, we have supplemented our study with experiments on ResNet (**Appendix I, Fig. 17**), and across these architectures, our proposed metrics remain strongly correlated with alignment scores, demonstrating robustness to model class and pretraining objectives.
>
> > Q3. Concern about purely correlational analysis.
>
> In response, we added a causal-style manipulation experiment (**Appendix O, Fig. 23**). For the ten models with the lowest alignment, we directly modified their singular value spectra to more closely follow a power-law distribution, thereby increasing the degree of scale-invariance in their embeddings. As the strength of this manipulation increased (**Fig. 23B**), alignment scores improved systematically. This manipulation primarily affected the stability of dimensionality across scales (**Fig. 23E**), supporting our interpretation.
>
> > Q4. Interpretation of the dimensional-stability slope.
>
> Following your suggestion, we removed the two largest scales (K = 8000 and 10,000) and recomputed the slopes. The correlations remain significant, indicating that the results are not driven by boundary effects (**Appendix J, Fig. 18A**).
>
> To further ensure robustness, we repeated the analysis with a reduced sample size of 1000, obtaining consistent results (**Appendix J, Fig. 18B**).
>
> Finally, we tested two additional dimensionality estimators, and the findings again remained stable (**Appendix J, Fig. 18C–D**).
>
> > Q5. Identical correlation values for PPA and RSC in Fig. 9.
>
> We have rechecked both the code and the results and confirmed no copy–paste or implementation errors. While the values for PPA and RSC are close, they are not identical: models perform slightly better on PPA than on RSC. This similarity arises because the ranking of models’ alignment performance is highly stable across higher-level visual regions. A model that performs well in one region typically performs well in another, leading to similar but not identical correlation patterns.
>
> > Q6. Low alignment in early visual regions.
>
> First, prior work showing strong performance in early visual cortex used models specifically trained to predict neural responses [1]. In contrast, our models were trained purely on image or vision–language tasks and have never been exposed to neural data. Our goal is to investigate whether AI models trained under general paradigms develop representations that resemble the information encoding patterns observed in the brain.
>
> Second, our evaluation metric differs. St-Yves et al. define prediction accuracy as “the Pearson correlation between predicted brain activity and measured brain activity on a single-trial basis” (page 13, Section “Cross-validated prediction accuracy”). In contrast, we compute the coefficient of determination on held-out test data using ridge regression fits.
>
> > Q7. Unclear intuitive notion of “scale.”
>
> We have added an intuitive explanation to clarify this point:
> “Larger values of K incorporate progressively more global structure, meaning that we start to include points that are farther away, revealing broader patterns and relationships. To put it simply, as we increase K, we're zooming out and looking at the neighborhood around the anchor from a wider perspective. In all analyses, we vary K over a prescribed range of scales and report how statistics evolve with K.”
>  (This text appears in Section 3.2 of the revised manuscript.)
>
> **Reference**
>
> [1] St-Yves, Ghislain, et al. "Brain-optimized deep neural network models of human visual areas learn non-hierarchical representations." Nature communications 14.1 (2023): 3329.

---

### Author Response · Authors · 2025-11-25
**Global Response**

To address all reviewer concerns comprehensively, we conducted a series of additional experiments and conceptual clarifications across architectures, evaluation protocols and theoretical interpretations.

**1.Additional Experiments on Architecture Robustness (qqDn(q2), XA1n(q1), Eohs(q2))**
To ensure generality, we supplemented our analyses with ResNet models (**Appendix I, Fig. 17**), confirming that our scale-invariance metrics consistently predict alignment across architectures.

**2.Causal-style Manipulation of Embedding Spectra  (qqDn(q3), UeYU(q3))**
We performed singular value spectrum manipulations on low-alignment models (**Appendix O, Fig. 23**), increasing scale-invariance and observing systematic improvements in alignment, demonstrating that embedding geometry causally contributes to alignment rather than correlations being purely incidental.

**3.Dimensional Stability and Slope Analysis  (qqDn(q4))**
We recomputed dimensionality slopes excluding boundary scales, varied neighborhood sample size, and tested alternative dimension estimators (**Appendix J, Fig. 18**), confirming that the results are robust to methodological choices and not driven by artifacts.

**4.Distinguishing Trivial vs. Non-Trivial Scale-Invariance (UeYU(q1))**
Experiments on uniform and Gaussian data (**Appendix M, Fig. 21**) show large scale-dependent changes in dimensionality, which helps distinguish trivial from non-trivial scale-invariance.

**5.Power-Law Spectra and Neural Data (UeYU(q3))**
We analyzed the power-law exponents and the log–log linear fit R^2 of covariance eigenvalue spectra (**Appendix L, Fig. 20**), showing that although these features are significantly correlated with alignment performance, their predictive strength is consistently weaker than that of our proposed scale-invariance metrics.

**6.Raw Image Scale-Invariance Control (XA1n(q4))**
We verified that the scale-invariance of raw stimulus images is not significantly correlated with alignment performance (**Appendix K, Fig. 19**), confirming that alignment is determined primarily by the embedding representations rather than the images themselves.

**7. Brain Region Specificity (Eohs(q4))**
Whole-brain analyses (**Appendix N, Fig. 22**) show no region consistently exhibits higher scale-invariance, indicating that scale-invariance is a global property rather than region-specific.

**8. Clarifications and Conceptual Refinements (Eohs(q2, q3))**
We clarified that our notion of scale-invariance refers to geometric scale-invariance in embeddings rather than perceptual invariance, and emphasized that it is a data-driven property rather than architecture-inherent, providing a mathematically grounded framework to interpret embedding geometry and neural alignment.

---

### Author Response · Authors · 2025-12-03
**Summary**

We sincerely thank the reviewers for their constructive feedback, which helped us substantially strengthen the paper.

**Core Contributions of the Paper**

* We identify geometric scale-invariance as a fundamental embedding-level property that reliably predicts the alignment performance of individual models. This shifts the focus from architecture- or task-level factors to a concrete geometric characteristic of the representations themselves.

* We validate the predictive power of our metrics across a large pool of pretrained models, multiple subjects, and multiple cortical regions, demonstrating that the effect is strong, stable, and neuroscientifically generalizable.

* We show that larger pretraining datasets consistently increase embedding scale-invariance and improve alignment, while finetuning disrupts this structure and reduces alignment. This reveals a direct, mechanistic pathway through which training regimes shape biological fidelity.

**Key Improvements Made During Rebuttal**

**1. Architecture Generalization**: Added full analyses on ResNet, confirming that scale-invariance predicts alignment across architectures.

**2. Causal Evidence**: Manipulated singular value spectra to artificially enhance scale-invariance, producing systematic alignment gains and addressing correlation–causation concerns.

**3. Comparison With Other Metrics**: Verified that power-law exponents/log–log R² correlate with alignment but are consistently weaker predictors than our metrics.


These additions make the paper more rigorous, more general, and conceptually clearer, reinforcing that geometric scale-invariance is a robust and mechanistically meaningful explanation for AI–brain alignment.

---

### Meta-Review · Area_Chair_QUUo · 2026-01-09

**Summary:**

Reviewer qqDn

(1) Limited discussion of prior work on measuring alignment between task-optimized neural networks and brains; (2) the set of models tested (60 ConvNeXt/CLIP variants) are narrow in the sense of sharing the same architecture (ConvNeXt) and being trained on the same dataset (ImageNet) with a supervised objective; (3) the analysis is correlational, but the claims are causal ("drives"); (4) Figure 4A suggests that the relationship between intrinsic dimensionality and alignment may be confounded by artefacts in the analysis; (5) there seem to be numerical errors in Figure 9; (6) the work fails to reproduce alignment scores reported in prior work; (7) scale is never intuitively defined; (8) relevant work on scale restructure visual representations, covariance spectra, and manifold smoothness are not cited.

Reviewer XA1n

(1) Same as Reviewer qqDn (2); (2) no mechanism for scale invariance is evidenced; like Reviewer qqDn (3), the measures are correlational.

Reviewer UeYU

(1) The authors' measure of scale and variance in neural representations captures trivial scale invariance. The more proper measure is power law covariance spectra of the covariance matrix of neural representations. The authors seem not to understand these results; (2) the authors do not justify their claim that traditional methods such as those in (1) "struggle" with the datasets tested here.

Reviewer Eohs

As the reviewer themselves states: (1) "Computational and neuro-scientific justifications of the results are very limited"; (2) "Justifications about model selection are missing"; (3) "The definition of 'scale-invariance' is different from common understanding, potentially misleading."

**Reviewer Concerns:**

##### Addressed

###### Reviewer qqDn (1), (8)

Requests to add references to relevant work could be added in a revision.

##### Outstanding

###### Reviewer qqDn (2); Reviewer XA1n (1)

This is responded to by the inclusion of around 20 residual networks trained on ImageNet in Appendix I. The authors claim this set is "diverse," but they differ only in the number of layers, as they will share the same architectural motif, and are pre-trained on the same dataset. Given the manifold of large-scale models with differing architectures, pretraining datasets, and objectives, this is a limited demonstration.

###### Reviewer UeYU (1), (2); Reviewer Eohs (3)

The authors address part of these concerns by reporting the more classical measure suggested by Reviewer UeYU (covariance spectrum). However, the authors make several unsubstantiated claims to refute its use over their method: "they do not fully reflect the multi-scale geometric structure." What is meant by multi-scale geometric structure *in the context of brain models*, and and in particular what is desirable about it for computational modelling of the brain (also discussed below), is insufficiently clear.

This claim introduced in the revision: "While these methods are effective at detecting whether a system exhibits scale-invariant behavior, they provide limited insight into the degree to which a system approaches true scale-invariance." What is "true scale invariance"? Like the previous claim that "these [classical] methods [to describe scale invariance] often struggle with discrete datasets and provide limited ways to quantify the degree of scale-invariance," this claim is unsubstantiated.

###### Reviewer XA1n (2); Reviewer qqDn (3)

The points about the approach being correlational remain unaddressed. Interventions like dataset size and finetuning are shown, as well as a toy manipulation on the covariance spectrum; these factors appear to jointly affect geometric scale invariance and neural predictivity, but do not isolate scale invariance as the cause rather than a common effect. Further, the mechanistic relationship between geometric scale invariance and better brain models is not directly tested. The authors should aim to directly answer the question: What does having a _predictor_ of neural predictivity matter, if we do not understand _why_ the two are related? The authors do not yet answer this (nor the related question in Figure 1, "Which embedding property can *explain* alignment?" emphasis mine).

**Reviewer Scores:**

Reviewer UeYU brought forth a critical claim about the triviality of the scale-invariance measure. I do not think that the authors' response would have addressed this concern, as they confirm that their scale-invariance measure detects trivially scale-free structure, and they require another "neighborhood structure component" to complement their scale-invariance measure. The authors should focus on getting the foundations right by engaging with the classical literature on scale-free systems, including the citation that the reviewer gives. This, in addition to the lack of acknowledgement of the focus on correlational results (Reviewer XA1n (2); Reviewer qqDn (3)), weakens the paper enough for me to hesitate recommending it for acceptance, even if the preliminary empirical findings may point to something interesting, as Reviewer UeYU themselves notes.

---

### Decision · Program_Chairs · 2026-01-26

Reject